# Leveraging spatial-angular redundancy for self-supervised denoising of 3D fluorescence imaging without temporal dependency

Zhi Lu [1,2,3,4,5,6,11], Wentao Chen[3,7,11], Feihao Sun [2,3,4], Jiaqi Fan [3,8], Xinyang Li [4,9], Zhenqi Fu [2,3,4], Manchang Jin [3,6,7], Jiamin Wu [2,3,4,10] & Qionghai Dai [2,3,4,10]

Photon noise is one of the fundamental limits in fluorescence imaging. Despite broad applications, existing self-supervised denoising methods rely on either temporal redundancy or spatial redundancy, leading to degradation in either temporal resolution or spatial resolution, especially for 3D imaging. Here, we propose light field denoising (LF-denoising), a self-supervised transformer framework leveraging spatial-angular redundancy based on the high-dimensional light field measurements to achieve high-fidelity denoising without relying on temporal information and avoiding associated artefacts in spatial domain. We demonstrate the advantage of LF-denoising over previous methods in highly dynamic 3D imaging with both simulations and experimental data across different species. Combined with state-of-the-art light field microscopy variants, we achieve long-term high-speed high-resolution 3D intravital imaging on diverse animals including zebrafish, *Drosophila* and mice, with ultra-low excitation power of 10 μW/mm². Specifically, we show that LF-denoising well preserved the temporal causality with superior denoising performance, which is critical for quantitative biology analysis in immunology and neuroscience.

High-fidelity subcellular 3D imaging in live organisms with minimal phototoxicity is crucial for studying multicellular behaviors and functions[1–4]. However, photon noise is one of the fundamental limits, leading to fragmented images and artefacts[5]. For example, prolonged 3D imaging sessions often encounter significant shot noise with low signal-to-noise ratios (SNR), especially when imaging photosensitive tissues or cells over extended periods.

Deep learning-based denoising methods have shown promise in enhancing noisy microscopy images[6,7]. Nevertheless, existing methods face several bottlenecks, especially for 3D imaging. Supervised learning-based networks[8,9] can substantially suppress image noise by acquiring low- and high-SNR data pairs simultaneously; however, required targets are barely accessible for light-sensitive samples[10] or fast, nonrepetitive 3D biological activities[11,12]. Consequently, self-supervised networks are more commonly used, denoising directly on low-SNR data by relying on either temporal or spatial redundancy. On one hand, DeepInterpolation[6] and DeepCAD-RT[7] rely on temporal redundancy, in which signals from consecutive frames are regarded as almost the same and noises are considered to be independent. The temporal dependency may cause motion blur when imaging highly

¹Department of Psychological and Cognitive Sciences, Tsinghua University, Beijing, China. ²Department of Automation, Tsinghua University, Beijing, China. ³Institute of Brain and Cognitive Sciences, Tsinghua University, Beijing, China. ⁴IDG/McGovern Institute for Brain Research, Tsinghua University, Beijing, China. ⁵Zhejiang Hehu Technology, Hangzhou, China. ⁶Hangzhou Zhuoxi Institute of Brain and Intelligence, Hangzhou, China. ⁷School of Information Science and Technology, Fudan University, Shanghai, China. ⁸Tsinghua Shenzhen International Graduate School, Tsinghua University, Shenzhen, China. ⁹College of AI, Tsinghua University, Beijing, China. ¹⁰Beijing National Research Center for Information Science and Technology, Tsinghua University, Beijing, China. ¹¹These authors contributed equally: Zhi Lu, Wentao Chen. ✉e-mail: wujiamin@tsinghua.edu.cn; qhdai@tsinghua.edu.cn

dynamic 3D samples, limiting their application to a relatively small range, such as functional calcium dynamics in which neuronal structures are kept unchanged[6,7] or in vitro cultured living cells that move slowly[13]. On the other hand, spatial redundancy can be utilized for denoising through spatial oversampling, with typical methods such as SRDTrans[14] and SN2N[15]. But for fast volumetric imaging, spatial oversampling is not easily achieved and comes at the cost of a reduced field of view or lowered resolution. Therefore, both temporal and spatial redundancies are often insufficient, particularly in high-dynamic, orchestrated 3D samples, resulting in degradation in either temporal or spatial resolution. Exploring higher-dimensional redundancy for 3D image denoising is imperative.

Light field microscopy (LFM) provides a high-dimensional sampling that captures both spatial and angular information, offering spatial-angular redundancy to achieve sophisticated 3D imaging with high spatiotemporal resolution[16–27]. The core advantage of LFM lies in its highly parallelized 3D volumetric imaging capability. Traditional confocal[28] and two-photon microscopes[29] require slow, point-by-point scanning across the 3D volume to achieve volumetric imaging. More recently, light-sheet microscopy[30] has improved the physical scanning mechanism, yet it still relies on sequential plane-by-plane scanning, which remains relatively slow and is further constrained by sample transparency. In contrast, LFM leverages high-dimensional spatial-angular information to enable single-shot 3D volumetric imaging with a single camera exposure. The unique capability has greatly advanced large-scale, high-speed biological applications and is now becoming commercially available and widely used by the scientific community[31–33]. Compared to exploiting temporal or spatial redundancy in a single dimension, the joint spatial-angular redundancy inherent in LFM holds the potential for enhanced denoising performance. However, it has not been fully leveraged by previous methods due to the ill consideration of the complex spatial-angular model. Fixed pattern artefacts and under-performed effects occur when using existing self-supervised networks. Until now, denoising LFM images still relies on traditional methods, such as block matching[34], total variation regularization[35] or dictionary learning[36], all of which easily overestimate noise and degrade performance. Deep learning-based light field denoising is constrained to macroscopic-scale scenes for natural images without wave-optics modeling[37,38]. Therefore, developing denoising methods by leveraging spatial-angular redundancy is urgently needed to achieve long-duration intravital 3D subcellular imaging under low-light conditions.

To address such problems, we develop a self-supervised learning-based imaging framework termed LF-denoising based on the high-dimensional light field measurements, to push 3D fluorescence imaging beyond the shot noise limit. The proposed LF-denoising comprehensively considers the attribute of spatial-angular redundancy in LFM variants, to disentangle underlying signals from noisy recordings[39–41]. Implementing the bidirectional angular traversing strategy, we created four sets of high-dimensional data pairs in the epipolar plane image (EPI) domain, whereby spatial-angular continuity and redundancy can be fully investigated[42]. Equipped with two concurrent transformers, LF-denoising effectively removes noise in spatial-angular images without compromising resolution. The consistent relationship between multiple spatial-angular components ensures accurate volume reconstruction. We demonstrated the versatility of LF-denoising across various spatial-angular imaging setups (conventional LFM, high-resolution scanning LFM (sLFM), two-photon synthetic aperture microscopy (2pSAM)), diverse animals (zebrafish, *Drosophila*, mice) and multiple research fields (hemodynamics, developmental biology, immunology, neuroscience). Quantitative analysis characterized that LF-denoising improved SNR by over 11 dB, and resolved 3D subcellular activities over 10 h at unprecedentedly low excitation light levels of 10 μW/mm². Moreover, LF-denoising accurately preserved event causality in neural recordings without temporal dependence, which may be compromised in previous methods.

## Results

### Principle and benchmark of LF-denoising

Inspired by studies utilizing Noise2Noise for image denoising[7,39], a natural idea is to apply self-supervised denoising on LFM data after 3D reconstruction. However, deconvolution alters the distribution of signal and noise and fails to account for spatial-angular correlations, resulting in poor performance (Supplementary Fig. 1). Therefore, it is necessary to perform denoising directly on the high-dimensional data acquired by LFM. Considering that 4D spatial-angular correlation is concealed within LFM, traditional temporal or spatial redundancy is insufficient. We explore a strategy to extract clear signals from complex light field acquisitions by utilizing high-dimensional spatial-angular redundancy (Fig. 1a). Specifically, we used the EPI format for denoising, as it explicitly represents these spatial-angular constraints[42]. By employing bidirectional angular traversals, noisy EPI pairs can be established at different axes. Along with noisy spatial-angular images, we obtain three groups of noisy data pairs for self-supervised training (Fig. 1b). LF-denoising features a hybrid architecture with two transformers to leverage surrounding spatial-angular pixels for improving SNR of EPI, an attention-based fusion module to transform EPI pixels into spatial-angular space, and a global connection with multiple orthogonal masks to enhance fidelity (Supplementary Fig. 2). The bidirectional structure effectively preserves spatial-angular correlations to present high-resolution details while maintaining fidelity. Removing part of the network degrades performance (Supplementary Fig. 3a–d). To prevent overfitting to sample textures in the training set, LF-denoising is designed to be as compact as possible with only 18 million parameters. Data augmentation before training further reduces overfitting and enhances performance (Supplementary Fig. 3e, f). During inference, noisy spatial-angular images are directly converted into clean ones, achieving an 11 dB SNR enhancement without data supervision and temporal dependency (Fig. 1c–e). For efficiency, 81 central angular views were utilized to achieve desirable denoising performance with accelerated network training and inference (Supplementary Fig. 4).

We validated LF-denoising through quantitative experiments. We compared it with state-of-the-art methods, including analytic methods (block matching and 3D filtering (BM3D[34])) and deep learning networks (Noise2Void[41], Noise2Noise[39], DeepInterpolation[6], DeepCAD-RT[7], DeepSeMi[13], SRDTrans[14]). Fully supervised networks like CARE[8] were excluded due to the inaccessibility of high-SNR ground truth in microscopy. We benchmarked LF-denoising using data from sLFM[25], a representative LFM variant. We simulated the sLFM imaging process with low photons on a 3D confocal stack and enhanced it using LF-denoising (Fig. 2a). BM3D results were extremely blurry, failing to distinguish signal from noise (Fig. 2b). Learning-based methods, while effective in noise removal, resulted in some detail loss. Methods like Noise2Void and Noise2Noise, crafted for natural images, tended to reduce high-frequency components at the submicron level. DeepInterpolation, DeepCAD-RT and DeepSeMi engineered for denoising fluorescence images, showed compromised fidelity when applied to spatial-angular data instead of spatiotemporal data due to temporal dependency. Due to the substantial variation across different angular views in light field images, the signals corresponding to consecutive frames in this transformed space also become dissimilar, which leads to a reduction in resolution. SRDTrans, which leverages spatial redundancy, is less dependent on temporal continuity but still struggles with single-frame data, resulting in reduced image contrast due to its continued use of spatiotemporal convolution in the denoising process. These methods focused primarily on spatial information, disrupting angular correlations and undermining reconstruction quality (Fig. 2c). Differently, by addressing spatial-angular constraints in LFM, LF-denoising preserved the resolvability of tiny structures, which are mistaken by conventional methods. Quantitatively, LF-denoising achieved optimal denoising performance with a 3 dB

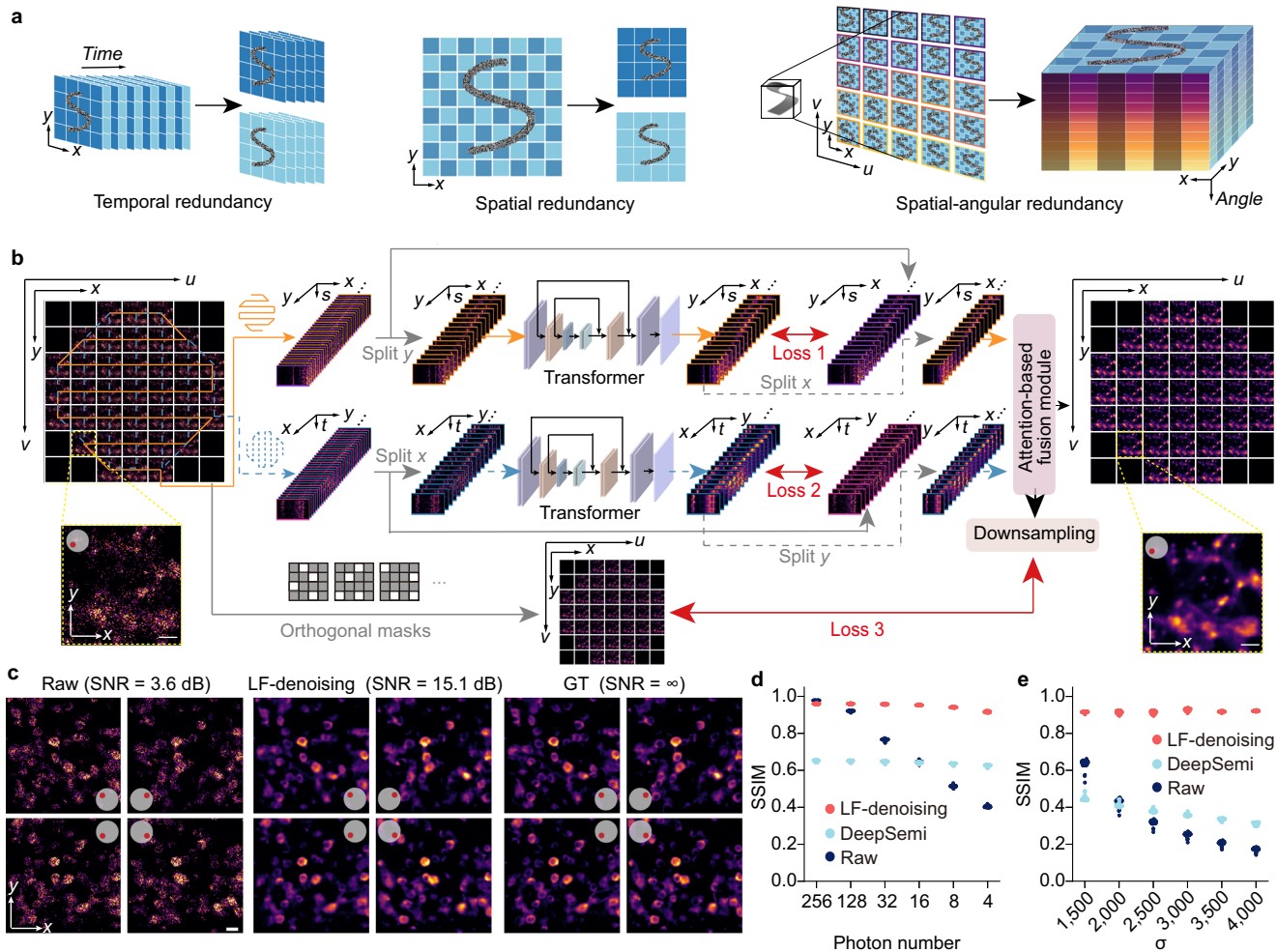

**Fig. 1 | Principle of LF-denoising. a** Comparisons between strategies of self-supervised denoising by relying on temporal, spatial, and spatial-angular redundancy. **b** Schematic of the LF-denoising processing pipeline. Noisy spatial-angular images are first converted into two EPIs via bidirectional angular traversing. These noisy EPIs are divided into four sub-stacks along the $x$ and $y$ dimensions, serving as inputs and targets for training transformer-based networks in a self-supervised manner. An attention-based module then fuses the two channels to produce clean spatial-angular images. Multiple orthogonal masks and downsampling operators are employed to establish a global loss function, enhancing fidelity. Loss 1 and Loss 2 are sub-network loss functions defined on the EPI representation, whereas Loss 3 is a global loss function defined on fused spatial-angular representation. **c** Spatial-angular images of simulated neuron data before and after LF-denoising, with clean images provided as ground truth. SNR values are indicated above each image. LF-denoising performance indicated by the SSIM across varying photon numbers (**d**) and Gaussian standard deviations (**e**), with distinct improvement over state-of-the-art methods. Image bit depth is set to 16. Scale bars: 20 μm (**b**, **c**).

improvement in peak signal-to-noise ratio (PSNR) and a 30% reduction in root-mean-square error (RMSE) compared to state-of-the-art algorithms (Fig. 2d). Simulated tubulins and bubbles from the bubtub dataset[43] exhibited similar enhancement with LF-denoising (Supplementary Fig. 5). Even as noise increased, LF-denoising consistently outperformed conventional methods, regardless of whether the noise followed Poisson (Fig. 1d and Supplementary Fig. 6) or Gaussian distributions (Fig. 1e and Supplementary Fig. 7).

### Temporal independency enhances sensitivity of LFM for highly dynamic 3D recordings

Observing cells and organelles in living animals requires low excitation power to prevent photodamage[44]. The accompanying shot noise would contaminate the signals, complicating the measurements. While previous denoising techniques have advanced in handling videos with slow sample motions, processing data from highly motile bioprocesses remains extremely challenging (Supplementary Fig. 8). Although LFM can record volumetric signals within a snapshot for high-speed 3D observation over a long period[17], there has been no dedicated

algorithm designed to recover clean dynamics in LFM recordings. For example, the rapid movement of cells within the animal's circulatory system poses a challenge for learning-based video denoising. Methods relying on spatial redundancy also fall short due to insufficient spatial sampling rate, difficult to distinguish fine structures with high resolution.

To demonstrate the advantages of our method, we applied LF-denoising to enhance data on flowing blood cells in beating hearts of zebrafish larvae captured by LFM. Under low excitation power, the raw measurements were severely affected by noise (Fig. 3a). The data were imaged at a high speed of 50 volumes per second (VPS), resulting in substantial differences between frames. Methods like DeepCAD-RT and DeepSeMi, which require temporal redundancy that assumes consecutive frames can be treated as independent samples of the same signals or require surrounding pixels in the spatiotemporal domain for denoising, were incapable of handling such dynamic data. Consequently, these methods produced results with motion blur and haze (Fig. 3b, c). Although SRDTrans reduced reliance on temporal continuity, it required spatial redundancy, which is not present in LFM,

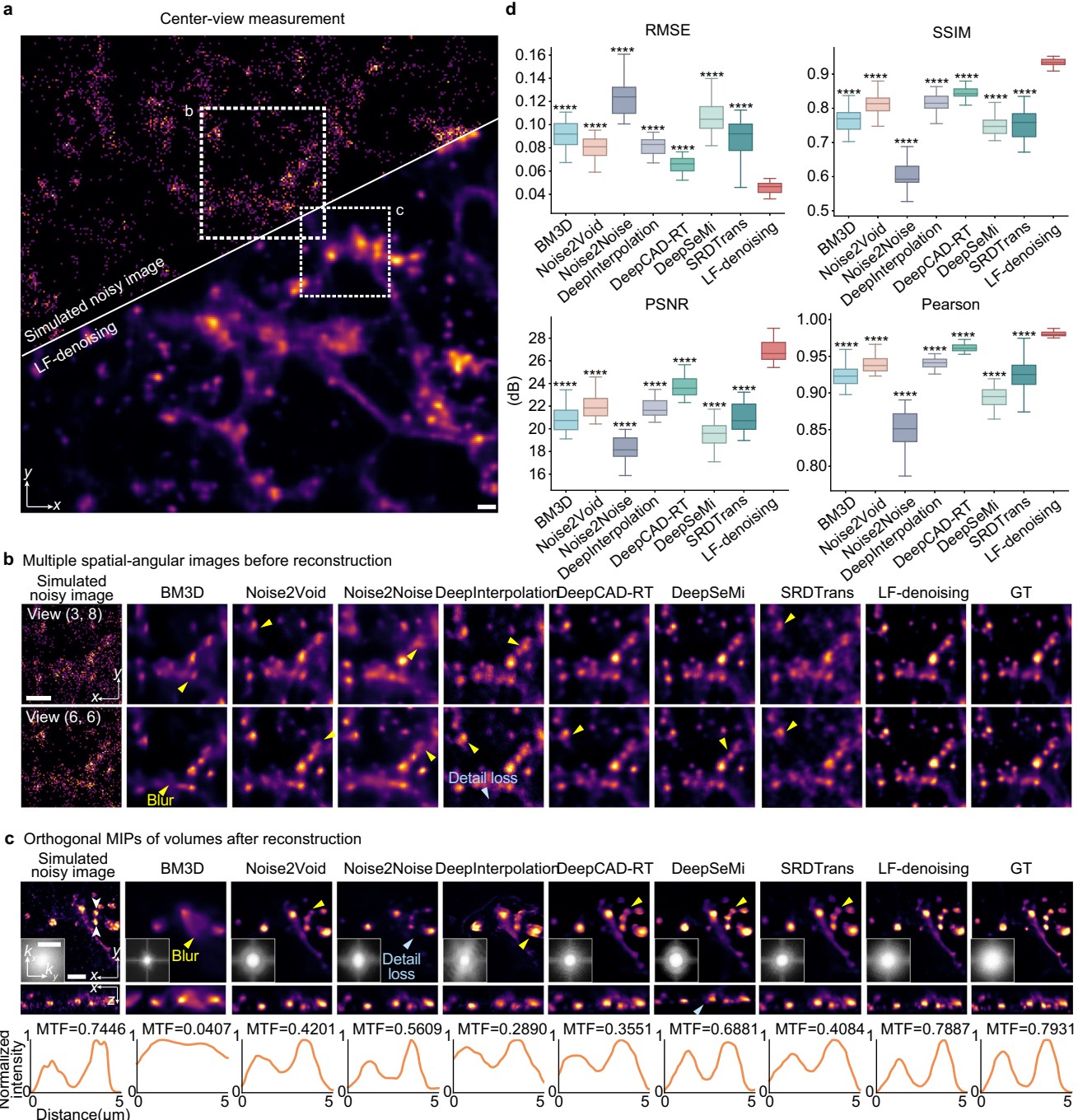

**Fig. 2 | Benchmark of LF-denoising with various denoising methods on the data captured by LFM. a** Center views of AICS cells labeled with microtubules, imaged using sLFM at low SNR in simulation, before and after LF-denoising. The image bit depth is set to 16, Gaussian noise variance to 5, and the photon number of maximum intensity to 100. **b** Enlarged regions obtained by different denoising methods, including BM3D, Noise2Void, Noise2Noise, DeepInterpolation, DeepCAD-RT, DeepSeMi, SRDTrans and LF-denoising. Two example angular views are shown. High-SNR data is regarded as ground truth. Yellow and blue arrows indicate the image blur and detail loss. **c** Another enlarged regions after reconstruction, with insets showing corresponding Fourier transforms. The normalized intensity profiles and modulation transfer functions (MTFs) of white arrows in the left most image across adjacent tiny structure are shown for comparison. The presence of a valley in the middle of the MTF curve indicates strong resolving capability of LF-denoising for spatially adjacent structures. **d** Boxplots showing comparisons

between different denoising methods using RMSE (lower is better), SSIM, PNSR and Pearson correlation (higher is better). Metrics are calculated on data after reconstruction. The boxplot formats: center line, median; box limits, lower and upper quartiles; whiskers, 1.5-fold interquartile range. All $P$ values were calculated using one-sided independent t test, significance at $P < 0.05$. $*P < 0.05$, $**P < 0.01$, $***P < 0.001$, $****P < 0.0001$. $n = 30$ denotes the number of individual samples. $P$ values are $1.71 \times 10^{-19}$, $9.03 \times 10^{-19}$, $7.35 \times 10^{-25}$, $3.55 \times 10^{-23}$, $2.71 \times 10^{-15}$, $3.51 \times 10^{-24}$ and $1.28 \times 10^{-11}$ from left to right for RMSE, and $2.95 \times 10^{-19}$, $9.84 \times 10^{-19}$, $6.98 \times 10^{-33}$, $2.48 \times 10^{-17}$, $4.72 \times 10^{-23}$, $2.38 \times 10^{-28}$ and $2.49 \times 10^{-12}$ from left to right for SSIM, and $9.02 \times 10^{-25}$, $1.35 \times 10^{-22}$, $6.85 \times 10^{-39}$, $3.02 \times 10^{-27}$, $8.39 \times 10^{-16}$, $3.29 \times 10^{-35}$ and $3.43 \times 10^{-17}$ from left to right for PSNR, and $3.76 \times 10^{-18}$, $2.51 \times 10^{-18}$, $1.89 \times 10^{-22}$, $1.63 \times 10^{-23}$, $8.43 \times 10^{-18}$, $5.41 \times 10^{-25}$ and $1.33 \times 10^{-10}$ from left to right for Pearson correlation. The asterisks indicate the significant levels of comparisons between each method and LF-denoising. Scale bars, 5 μm (**a**–**c**) and 2 μm⁻¹(**c**).

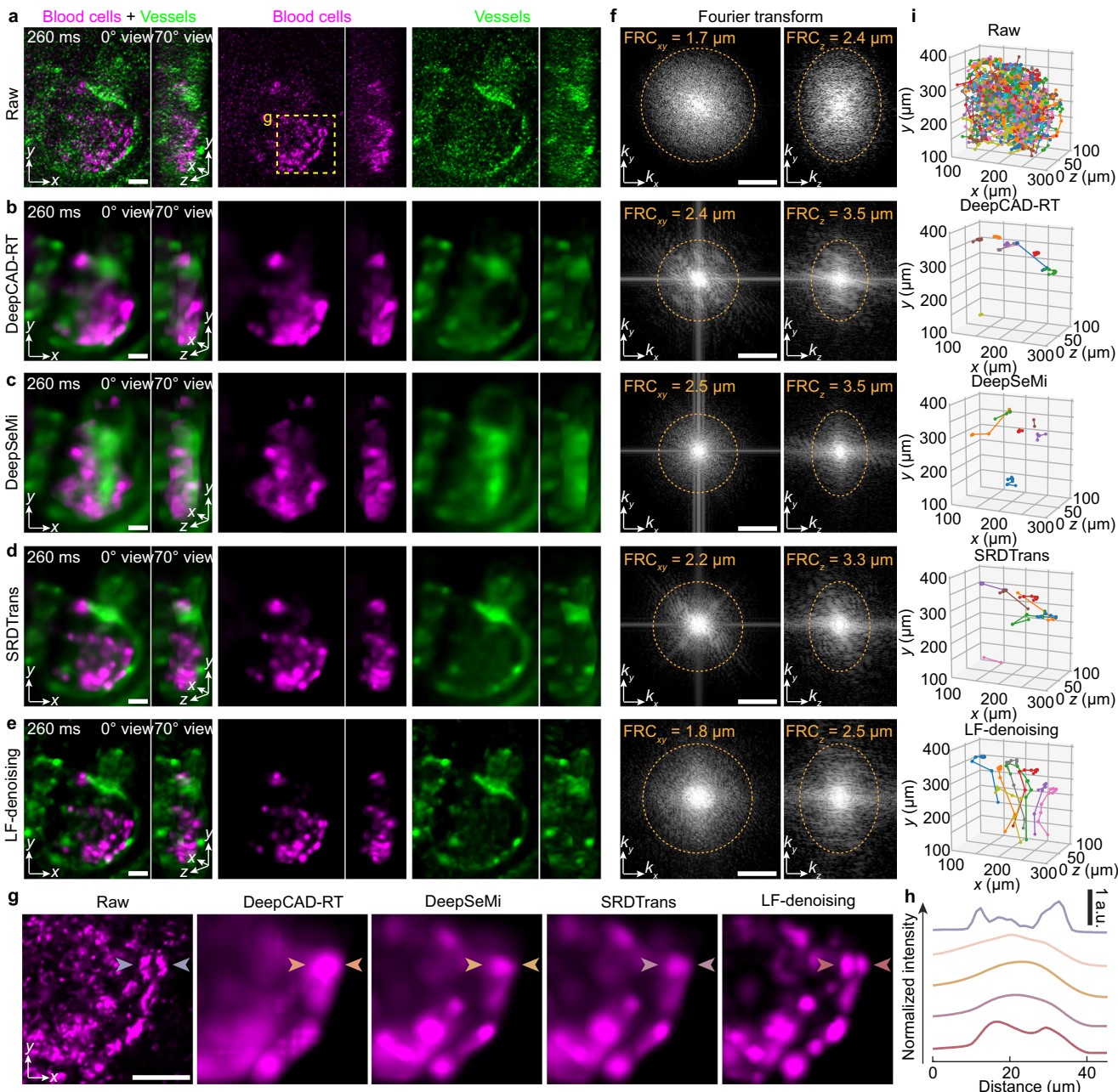

**Fig. 3 | LF-denoising shows superior performance in highly dynamic 3D imaging, such as heart beating in zebrafish larvae.** Maximum intensity projections (MIPs) of a beating zebrafish heart along different directions for: low-SNR raw recordings in (**a**) DeepCAD-RT enhanced recordings in (**b**), DeepSeMi enhanced recordings in (**c**) SRDTrans enhanced recordings in (**d**) and LF-denoising enhanced recordings in (**e**). All images are processed with 3D reconstruction. **f** Corresponding Fourier transforms of the vasculature channels with estimated resolutions determined by Fourier ring correlation. **g** Enlarged regions of the blood cell channel.

**h** Normalized intensity profiles along the lines indicated by arrows in (**g**) showing that adjacent cells, which were unresolved by previous algorithms, were distinguished by LF-denoising. **i** 3D tracking trajectories of representative blood cells flowing through the heart, obtained using different methods. Each trace represents the spatial position change at the center of each cell, and different colors represent different types of cells that have been identified. Scale bars: 50 μm (**a–e, g**) and 0.2 μm⁻¹ (**f**).

leading to underperformance (Fig. 3d). In contrast, LF-denoising leverages spatial-angular redundancy to accurately recover the real sample signal distribution without relying on temporal dependency. Blood cells and vascular structures were clearly resolved, as indicated by the extended high-frequency components[45] (Fig. 3e, f and Supplementary Movie 1). LF-denoising preserved high resolution, enabling the recognition of adjacent cells that were indistinguishable with conventional methods (Fig. 3g, h). Furthermore, only with LF-denoising enhanced recordings can we accurately track cell trajectories during cardiac cycles within 0.2 s. (Fig. 3i). Thus, the temporal

independence of LF-denoising extends its applications to highly dynamic bioprocesses.

## Spatial-angular redundancy facilitates high-fidelity denoising without artefacts in sLFM

We further integrated LF-denoising into sLFM for intravital subcellular observations. sLFM employs a microlens array (MLA) for light field modulation. However, the tens of thousands of lenslets on the MLA pose a challenge in ensuring uniformity during manufacturing[46]. This issue becomes more pronounced with limited photons, where

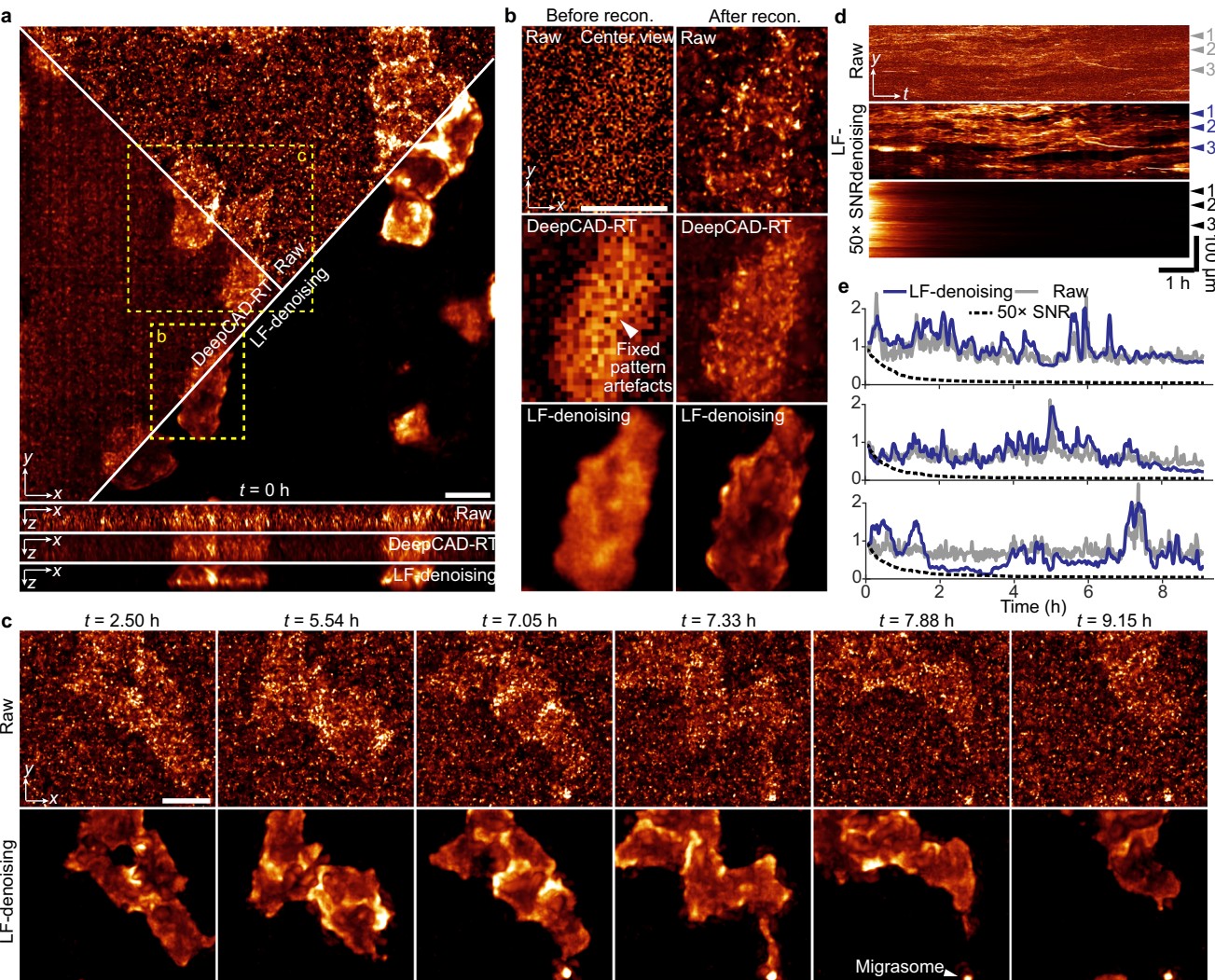

**Fig. 4 | LF-denoising with sLFM imaging enables day-long high-speed observation of 3D membrane dynamics with ultralow excitation power of 10 μW/ mm². a** Orthogonal MIPs of raw data and results processed by DeepCAD-RT and LF-denoising at *t* = 0 h. **b** Comparison of an enlarged region before and after 3D reconstruction using different methods. **c** Representative raw frames (upper row) and LF-denoising enhanced frames (lower row), illustrating the process of migrasome formation in vivo. **d** Kymographs along the *yt* plane of the low-SNR raw recording (upper row), LF-denoising enhanced recording (middle row), and synchronized high-SNR recording with 50-fold SNR. Kymographs were generated by calculating the MIP over the *x* and *z* axes. **e** Normalized intensity traces over time extracted from kymographs in (**d**). Scale bars: 20 μm (**a**–**c**).

nonuniform transmittance between microlenses can be exacerbated by previous methods (Supplementary Fig. 9). In high-SNR imaging, the raw measurement is clean, and denoising algorithms perform well. However, in low-SNR imaging, the noise values fluctuate significantly over time, while signals and MLA patterns remain constant. Due to its reliance on temporal information, DeepCAD-RT suffers from severe fixed pattern artefacts and fails to extract the signal correctly, resulting in outcomes similar to a direct average over time. LF-denoising, on the other hand, extracts and fuses information in spatial-angular domain instead of temporal domain, providing an effective solution to fixed pattern artefacts while achieving desirable denoising performance (Fig. 4a, b; Supplementary Fig. 10 and Supplementary Movie 2). With its high sensitivity, LF-denoising facilitated a 10 h 3D observation of migrasome formation in zebrafish embryos in vivo at an extremely low light intensity of 10 μW/mm² (Fig. 4c). Increasing the laser intensity to acquire clean images would lead to phototoxicity and photobleaching, swiftly eroding fluorescent signals and challenging the revelation of biological phenomena (Fig. 4d, e).

Next, we applied LF-denoising in mammalian in vivo observations. To validate our method, we developed a synchronized program within sLFM to capture low-SNR and high-SNR recordings simultaneously (Supplementary Fig. 11). We imaged neutrophils in the vessels of living mouse livers using inverted sLFM. At 20× lower laser intensity, LF-denoising successfully preserved the intact morphology of cells that were otherwise obscured by noise in spatial-angular images (Fig. 5a and Supplementary Movie 3). Quantitative analysis showed substantial improvement with LF-denoising compared to raw measurements (Fig. 5b, c). After denoising, we could observe the formation of migrasomes, previously obscured by noise in raw images (Fig. 5d). We counted the number of particles with diameters less than 2 μm over time. The raw images had many misidentifications, whereas the outcome of our method was fundamentally consistent with high-SNR recordings (Fig. 5e, f).

To visualize neural activity in the mouse cortex, we built an upright sLFM system with approximately 100 μm penetration depth in tissues. We recorded neural activities in an Ai148D mouse genetically expressing GCaMP6f indicators for both low and high SNR simultaneously at 30 VPS (Fig. 5g and Supplementary Movie 4). Due to noise, both the spatial footprints and temporal traces of neurons were severely contaminated (Fig. 5h). After LF-denoising, neurons became

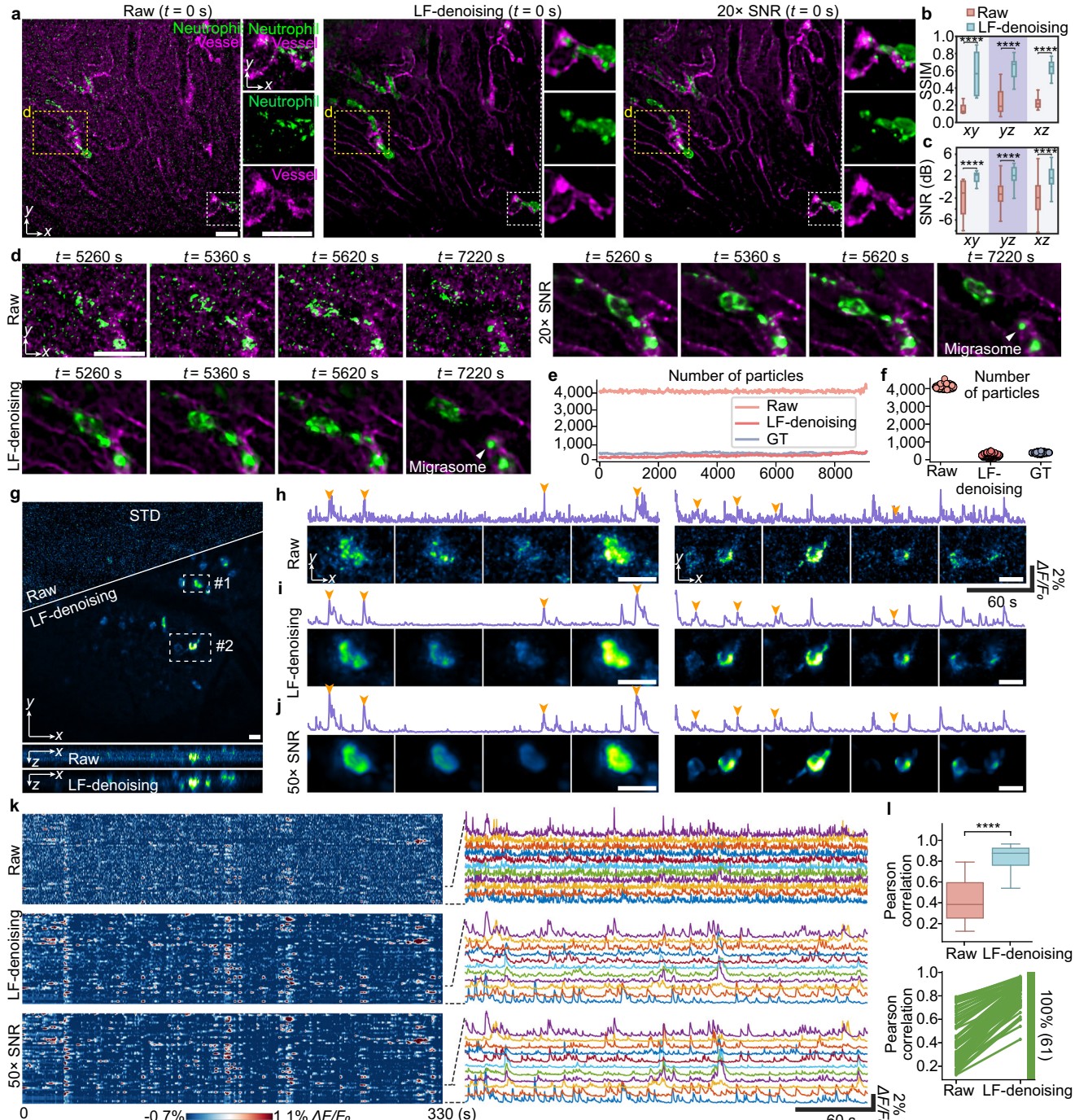

**Fig. 5 | LF-denoising reveals intravital subcellular behaviors of immune cells and neural activities in mice with high fidelity. a** MIPs of neutrophils (green) in vessels (magenta) of a living mouse liver, comparing low-SNR raw images (left), LF-denoising enhanced images (middle) and synchronized high-SNR recording with 20-fold SNR (right). Enlarged views are shown in the right. **b**, **c** SSIM and SNR indices of image slices along three dimensions before and after LF-denoising. The boxplot formats: center line, median; box limits, lower and upper quartiles; whiskers, 1.5-fold interquartile range. *xy* slice, *n* = 55; *yz* slice, *n* = 59; *xz* slice, *n* = 59. Asterisks represent significance levels tested with one-sided independent t test. $P = 2.4 \times 10^{-23}$ (*xy*), $6.2 \times 10^{-32}$ (*yz*) and $6.2 \times 10^{-51}$ (*xz*) for SSIM comparison; $P = 6.7 \times 10^{-14}$ (*xy*), $3.4 \times 10^{-7}$ (*yz*) and $2.9 \times 10^{-10}$ (*xz*) for SNR comparison. **d** Representative frames obtained by different methods, showing the process of migrasome generation in the mouse. **e** Counts of detected particles with diameters of less than 2 μm in the green channel over time. **f** Scatterplots visualizing the results in (**e**). *n* = 455

represents different frames. **g** Orthogonal MIPs in a mouse cortex labeled with GCaMP6f at 30 VPS, comparing low-SNR raw images and LF-denoising enhanced images. The displayed images show the standard deviation over time. Fluorescence traces and representative frames in two enlarged regions, for raw recordings in (**h**) LF-denoising enhanced recordings in (**i**) and synchronized high-SNR recordings with 50-fold SNR in (**j**). Images at four representative timestamps indicated by orange arrows are shown. **k** Fluorescence traces automatically extracted from results in (**g**). Zoom-in traces are shown in the right. **l** Boxplot showing Pearson correlation of fluorescence traces before and after LF-denoising. The boxplot formats: center line, median; box limits, lower and upper quartiles; whiskers, 1.5-fold interquartile range. *n* = 61 represents the number of traces. Each line represents one traces. Green color indicates increased Pearson correlation. Asterisks represent significance levels tested with one-sided independent t test. $P = 1.7 \times 10^{-24}$. Scale bars, 20 μm (**a**, **d**, **h**–**j**) and 50 μm (**g**).

distinguishable, and calcium traces were liberated from noise, aligning closely with high-SNR recordings (Fig. 5i–l). Thus, even without temporal information, LF-denoising enhances fidelity across spatiotemporal dimensions by leveraging spatial-angular redundancy, making it widely applicable to diverse biological studies.

### LF-denoising preserves causal relationships of neural activities in vivo for 2pSAM

Ensuring the causality of events is a significant challenge in denoising, especially in neuroscience, where the correlation between stimuli and neural responses is crucial[47]. Previous denoising methods[6,7] often use convolution operators on the temporal dimension, which can cause event misalignment and impede accurate neural analysis, potentially leading to distorted conclusions. In contrast, LF-denoising could be expected to achieve satisfactory denoising results in neural recordings with causality by replacing temporal dependency with leveraging spatial-angular redundancy.

To verify the causal fidelity of LF-denoising, we applied it to data captured by our previously developed 2pSAM[24], with a slight modification of the angular traversing (Fig. 6a). The system was also based on spatial-angular acquisition. We conducted an experiment on a living *Drosophila* brain under odor stimuli (Fig. 6b). LF-denoising emphasized spatial-angular redundancy and correlation within a single frame, achieving denoising on images (Fig. 6c) and functional traces (Fig. 6d). We identified and averaged neural spikes (Materials and methods). Compared to previous methods, LF-denoising produced more accurate spike morphology (Fig. 6e). DeepCAD-RT deteriorated temporal resolution and widened spikes, while SRDTrans partially reduced temporal dependency but with fluctuation due to ill consideration of high-dimensional redundancy. Quantitative analysis showed that our method's average spikes were closest to raw measurements (Fig. 6f). Furthermore, previous denoising techniques intertwined information across the temporal domain, distorting causal relationships. Our approach addressed this issue effectively. Response traces near the stimulus highlighted the drawbacks of DeepCAD-RT and SRDTrans, which led to premature or delayed responses and trace fluctuations, indicating limited fidelity (Fig. 6g). Only LF-denoising preserved causality and fidelity by eliminating time dependency. Enhanced traces showed the same trend as raw data but with reduced shot noise (Supplementary Movie 5).

To further demonstrate causal relationships, we embedded functional traces into a high-dimensional manifold for visualization (Fig. 6h and "Methods"). Each latent feature corresponded to neural activity after each odor release category. After denoising, the traces became more distinct. The starting point of the trace represented the initial state before odor release, and the closer it was, the greater the event causality in the neural response. Quantitative analysis showed that LF-denoising could shorten the distance between latent features by at least threefold compared to previous methods (Fig. 6h). With LF-denoising, the causality between multiple brain regions also improved, as indicated by Granger causality analysis (Fig. 6i–k). Together, our proposed LF-denoising maintains causality and has the potential for application in a wide range of high-fidelity biological observations.

## Discussion

In conclusion, we present LF-denoising, a computational denoising framework based on high-dimensional measurements by leveraging spatial-angular continuity and redundancy, enabling long-duration, low-phototoxicity, high-resolution, high-fidelity intravital volumetric imaging beyond the shot-noise limit. LF-denoising employs a self-supervised approach in high-dimensional space, eliminating the need for high-SNR data and temporal priors. The ingenious architecture allows LF-denoising to surpass state-of-the-art methods, particularly in removing fixed pattern artefacts and preserving event causality. By enhancing signal extraction through spatial-angular constrains, LF-

denoising demonstrates improved generalization across different structures (Supplementary Fig. 12). Extensive intravital experiments confirm its outstanding performance with minimal photon budget, especially in long-term, high-dynamic applications.

LF-denoising stands out as a unique method for 3D image denoising by specifically leveraging spatial-angular redundancy. On one hand, unlike DeepCAD-RT and DeepSeMi relying on temporal redundancy, LF-denoising eliminates the dependency on temporal information and specializes in high-dimensional LFM from design, making it applicable to highly dynamic 3D samples while preserving event causality. On the other hand, SRDTrans and SN2N investigate spatial redundancy to replace temporal redundancy for denoising, the latter of which employs a self-constrained learning process that further generalizes the Noise2Noise concept to remove noise with randomness and reduces the need for an infinite amount of data[15]. However, SN2N still fails to consider data redundancy in high-dimensional spaces, which makes it less effective than LF-denoising in the comparison on spatial-angular data (Supplementary Fig. 13). In addition, V2V3D[48] is a recently proposed method that relies on angular redundancy for denoising, but the lack of spatial redundancy makes it struggle to preserve low-intensity details in spatial-angular data (Supplementary Fig. 14). The results show that LF-denoising achieved higher resolution and better detail by extracting and integrating features across multiple spatial-angular components.

Recently, unsupervised diffusion models offer advantages for image denoising, primarily by eliminating the need for paired clean-noisy training data, leveraging instead the inherent structure within noisy observations themselves, making them particularly valuable for real-world applications[49]. However, these benefits come with notable drawbacks. The iterative nature of reverse diffusion necessitates hundreds to thousands of network evaluations per image, making inference orders of magnitude slower than single-pass discriminative models like LF-denoising, severely hindering real-time use. Performance of diffusion models is also highly sensitive to the choice of sampling parameters, requiring careful tuning to avoid artefacts or excessive smoothing. These hyperparameter adjustments pose challenges for biologists. In contrast, the proposed LF-denoising method offers a faster, more stable, and practical solution for denoising microscopy data.

To broaden the applicability of our method, we have shown that LF-denoising can be generally adapted to high-dimensional data captured from LFM, sLFM and 2pSAM. The performance of LF-denoising was evaluated on 2pSAM data with only 13 angular views. Further reducing angular views would likely degrade performance, as spatial-angular redundancy collapses toward purely spatial redundancy. LF-denoising does not presume any properties of the imaging system, including optical aberrations. The method has been verified to generalize across different spatial-angular imaging setups with varying optical aberrations. Nevertheless, severe aberrations that substantially disrupt angular redundancy may adversely affect the performance of LF-denoising, which can be addressed using computational adaptive optics. Furthermore, with the inherent light field nature unaffected, our denoising framework can be incorporated with other advanced technologies in the future, such as virtual scanning[27] for improving the spatial resolution, confocal detection[50] for removing background fluorescence, and multiscale model[22] for reducing tissue scattering. We envision that LF-denoising will serve as a practical, bio-friendly tool, enabling accurate 3D observation of subcellular behaviors and functions in vivo without the photodamage caused by intense excitation light.

## Methods
### Experimental setup and imaging conditions
The inverted sLFM system was based on the setup reported in our previous work[25], working on a commercial epifluorescence microscopy (Zeiss, Observer Z1) and an MLA (RPC Photonics MLA-S100-f21) attached to the image plane. A sCMOS camera (Andor Zyla 4.2 Plus)

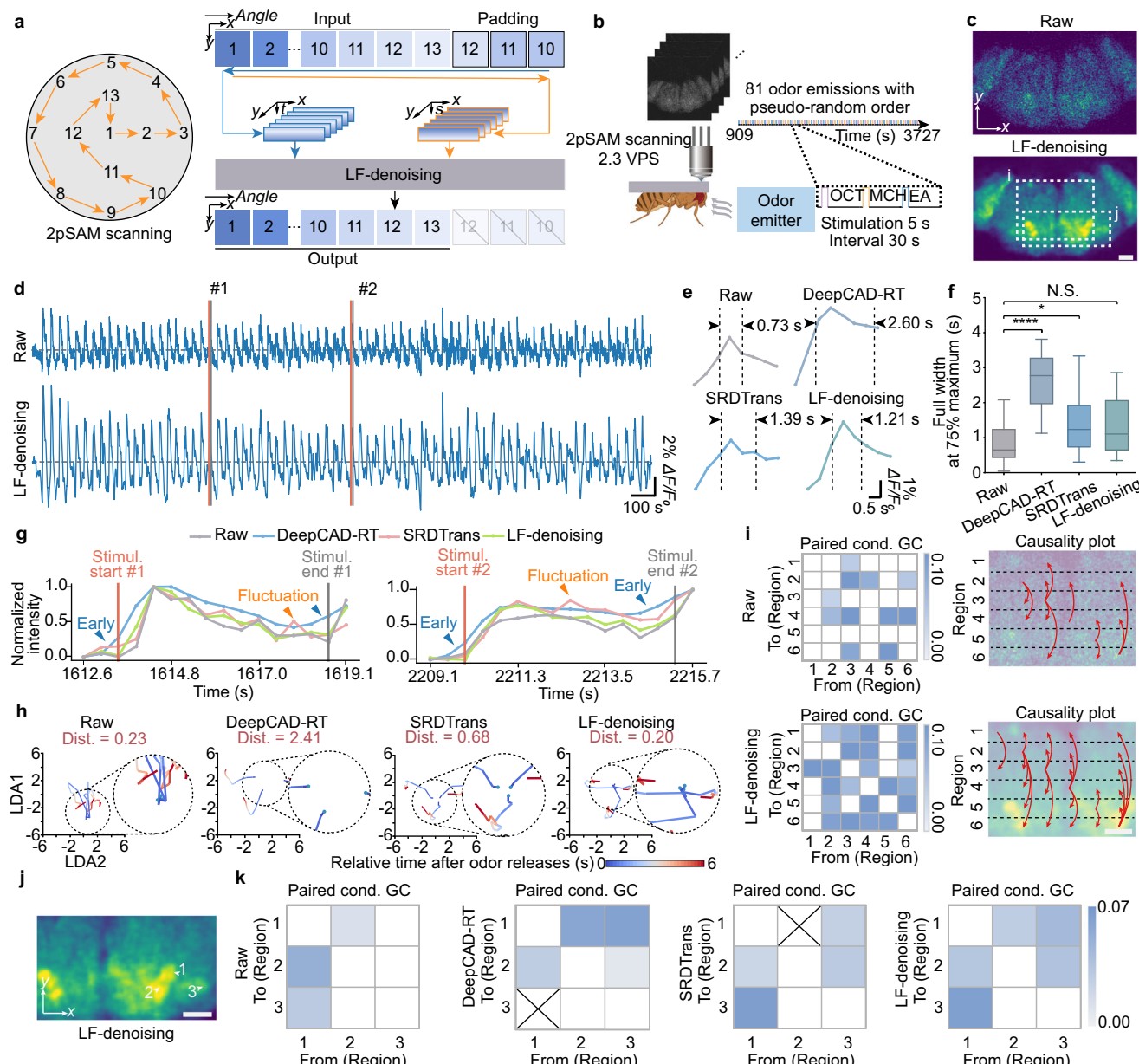

**Fig. 6 | LF-denoising preserves temporal causality for neural analysis with high fidelity. a** Left, system concept of 2pSAM with 13 scanning angles. Right, training strategy of LF-denoising on 2pSAM data. **b** Experiment on the *Drosophila* neural activity in response to odor stimuli. Created in BioRender. Fan, J. (2025) https://BioRender.com/cp0dmbd **c** Visualizations of a *Drosophila* brain expressing a genetically calcium indicator, comparing low-SNR raw image and LF-denoising enhanced image. **d** Functional traces extracted from raw images and LF-denoising enhanced images. **e** Average local peaks extracted from functional traces using different methods. **f** Boxplot showing full widths at 75% maximum of local peaks. The boxplot formats: center line, median; box limits, lower and upper quartiles; whiskers, 1.5-fold interquartile range. $n = 20$ represents the number of local peaks. Asterisks represent significance levels tested with two-sided independent t test. $P$ values are $5.36 \times 10^{-9}$, $4.22 \times 10^{-2}$ and $6.62 \times 10^{-2}$ from left to right. **g** Enlarged views showing the functional traces near the moment of odor stimuli. Arrows indicate errors induced by previous methods. **h** Average low-dimensional latent features of neural activity in olfactory brain regions after each category of odor release. Each trace represents the average odor response of an odor identity. Trace colors indicate the time relative to odor releases. Top, "Dist." denotes the maximum distance between the three start points and the average start point of all trials. Smaller distances indicate greater temporal accuracy. **i** Causality analysis on data before (upper row) and after LF-denoising (lower row). Left, matrix of Granger causality between each pair of six brain regions. Causality with $P < 0.05$ (Granger F test) is shown. Right, summary of causality between regions. Red arrows indicate inferred causality direction. **j** Three neurons in the olfactory brain region, shown in LF-denoising enhanced image. **k** Matrices of Granger causality between each pair of three neurons from raw data, DeepCAD-RT, SRDTrans and LF-denoising enhanced data. Causality with $P < 0.01$ (Granger F test) is shown. Cross denotes causality significant in raw data but lost after denoising. Scale bars, 40 μm (**c**, **i**, **j**).

deposited at the back focal plane of MLA, multichannel lasers (Coherent OBIS 405/488/561/640) providing excitation light and filtering module (Chroma ZT405/488/561/640rpcv2, ZET405/488/561/640xv2, ZET405/488/561/640mv2) installed in the filter cube of microscope, were used for multi-color imaging. Each microlens exactly covers 13 × 13 camera pixels with a designed 4f system with 0.845-fold

magnification. The 13 × 13 camera pixels can also be considered as the angular resolution of sLFM system. For the way of light field scanning, a 2D galvo system (Core Morrow P33.T2S) was inserted at the extended pupil plane and worked in a 3 × 3 scanning mode, where the amplitude of each drifting step was equal to one-third of the diameter of the microlens. A high-NA oil-immersion objective (Zeiss Plan-Apochromat

63×/1.4 Oil M27) is adopted in most biological experiments for high-resolution imaging. The results demonstrated in Figs. 1, 2, 4, 5a–f, Supplementary Figs. 1, 3–9, 12–14 and Supplementary Movies 1–3, were acquired by the inverted sLFM. When capturing zebrafish heart data in Fig. 3, we maintained the galvo stable. The sLFM was consequently downgraded to LFM.

The upright sLFM system was used for neural monitoring in vivo, by using optical components (Thorlabs Cerna series) and a customized MLA with a pitch size of 136.5 μm and a focal length of 2800 μm. A water-immersion objective (Olympus XLPLN25XWMP2) and a matched tube lens (Thorlabs TTL180-A) together consisted of the basic microscopic unit. A high-sensitivity sCMOS camera (Teledyne Photometrics Kinetix) was used for recording neural activities at high speed. Each microlens exactly covers $21 \times 21$ camera pixels. Other hardware devices, including lasers, filters and galvo, were identical to those in the inverted system. When acquiring scanning light field images, the galvo also worked on the $3 \times 3$ scanning mode. We also captured unscanned light field data to show the versatility of the proposed denoising method, during which the galvo remained unchanged at the offset voltage. The results demonstrated in Fig. 5g–l and Supplementary Movie 4, were acquired by the upright sLFM.

To capture synchronized low-SNR and high-SNR images, we wrote a hardware synchronization program based on the previous sLFM acquisition software (*sLFDriver*[26]). A time-division multiplexing method was used to trigger the lasers to turn on and off. The laser exposure durations for odd and even frames were set to different values (Supplementary Fig. 10). After rearranging the odd and even frames, the high-SNR data sequence and low-SNR data sequence can be acquired almost simultaneously. The low-SNR data were regarded as the input of denoising methods, while the high-SNR data served as the reference to evaluate the performance frame by frame. Both the upright and inverted sLFM systems utilized this synchronization control scheme.

We also acquired *Drosophila* data using 2pSAM. The 2pSAM system is consistent with what we previously described[24], in which a commercial femtosecond laser (Spectra-Physics InSight X3, Newport) was used for two-photon excitation. Angles at the conjugated objective plane are changed to scan the whole back pupil of the objective lens (25×/1.05 NA, water-immersion, XLPLN25XWMP2, Olympus). 13 illumination angles were used as previously stated[24].

Detailed imaging conditions and parameters used in biological experiments of this paper, including fluorescence label, laser, excitation power, exposure time, volume rate, objective lens and system setups, are illustrated in Supplementary Table 1. During imaging, relatively low laser intensities were applied to minimize photodamage in organisms.

## Overview of network architecture

In our proposed LF-denoising network, the input is a 4D tensor of low-SNR light field data, while the output is a 4D tensor of the high-SNR one. For the preprocessing of light field data captured by LFM or sLFM, we retained 81 central angular views (Fig. 1b). For light field data captured by 2pSAM, 13 views were padded to create 16 angular views (Fig. 6a). Subsequently, these spatial-angular images were rearranged into 3D tensors with *x-s* and *y-t* EPI representations (Fig. 1b). The tensors were then partitioned in spatial dimensions, yielding overlapping patches with a size of 128 pixels in both height and width. Specifically, *x-s* patches were split in the height dimension, generating two patches with the size of $64 \times 81 \times 128$ pixels, serving as an input-target pair of *x-s* sub-network. Simultaneously, the *y-t* patches were split in the width dimension for *y-t* sub-network. In addition, spatial random masks were applied to produce down-sampled patches used for the fusion module, to prevent the network from overfitting to artefacts generated during spatial downsampling processes. The masks group every four spatial

pixels of the input patch and randomly select two of them according to one of eight predefined patterns, resulting in two downsampled patches with half the original width and height. For angular dimension, the masks were identical across different views and always picked the same pixel. These masks were randomly generated in each training iteration, and the random strategy is consistent across different datasets.

An overview of LF-denoising network is given in Fig. 1b and detailed network structure is shown in Supplementary Fig. 2. For each sub-network, we emphasized spatial-angular redundancy for self-supervised image denoising. The input EPI patches were first fed into three convolution-based U-net encoders. Each encoder involved down-sampling with 3D maximum pooling layers (kernel size of $2 \times 2 \times 2$). The corresponding channel numbers for these encoders were 16, 32, and 64, respectively. Then the features were mapped into a representation using a 3D convolution layer (kernel size of $3 \times 3 \times 3$, and padding size of $1 \times 1 \times 1$), where the feature channel was usually set to 128. The resulting features were then sequentially passed through a temporal transformer and a spatial transformer. These layers worked in conjunction to leverage correlations across spatial and angular dimensions. Next, the features underwent convolution to be mapped back to feature representations with 64 channels, followed by U-net decoders that integrated them back into EPI patches using a 3D convolution (kernel size of $1 \times 1 \times 1$). The U-net convolutional encoders extract compact local features at low cost and reduce the dimensionality. The Transformer then captures long-range dependencies that encoders miss without the heavy computation at full resolution. Then, decoders recover spatial details with skip connections. This hybrid design is more computationally efficient than a purely Transformer model and preserves more local structure than a purely convolutional U-Net.

To mitigate information loss during the dataset preparation, we devised an attention-based fusion network that took the initial denoising outcomes from two sub-networks as input and produced a final denoising output. The ablation study demonstrated only single sub-network was insufficient and validated the effectiveness of employing two paths of network along with the fusion module (Supplementary Fig. 3). In our implementation, the output from *x-s* sub-network was split along the width dimension, while the output from *y-t* sub-network was split along the height dimension. Then the two features were rearranged into the same representation (in the form of angle × height × width, with the pixel size of $81 \times 64 \times 64$ or $16 \times 64 \times 64$, according to the input data). Subsequently, the two outputs were concatenated in an alternate order, forming the input for the fusion module. The fusion module also comprised two routes: residual swin transformer and convolutional block attention routes. In the former route, the input patch was first embedded with positional embedding, then processed through eight residual transformer blocks. Afterward, the output is fused using a double convolution layer along the angular dimension, resulting in 81 or 16 angular views. In the latter route, the input patch was initially passed into the double convolution layer, yielding 81 or 16 angular views. It is important to note that the parameters of the convolution layers were shared across both routes. Subsequently, the patch was processed by five convolution block attention modules. Each one consists of 3-channel attention layers and 3 spatial attention layers. The channel attention layers first calculated the spatial max pooling and spatial average pooling of the patches. These two pooled representations were then processed by a double convolution layer with shared weights. Finally, the output of channel attention was obtained by adding two convoluted features. The spatial attention layers first computed the channel-wise average and max of the patch while preserving the spatial dimensions. These two results were concatenated into a two-channel feature and then reduced into a single channel using a convolution layer. It is worth noting that the outputs of both channel attention layers and spatial

attention layers were element-wise multiplied with the input features to produce the final output features.

For network training, we typically used 1000 EPI patches of the same type of dataset, and it usually took approximately 20–25 epochs for convergence. For the two sub-networks, the pixel-wise mean absolute error (L1-norm error) and pixel-wise mean square error (L2-norm error) with equal weights were adopted as the loss function for each sub-network, which can be expressed as:

$$loss(X, Y) = 0.5 \times \|X - Y\|_1 + 0.5 \times \|X - Y\|_2, \quad (1)$$

where $X$ denotes the target of EPI patches and $Y$ denotes the output patches. For the fusion module, we introduced constraints between the fusion output and initial denoised fusion inputs from the two sub-networks. The whole loss function can be expressed as:

$$\begin{aligned} loss_{fusion} = {} & 0.4 \times [0.5 \times loss(Y_1, Y_f) + 0.5 \times loss(Y_2, Y_f)] \\ & + 0.6 \times [0.5 \times loss(Y_{m1}, Y_f) + 0.5 \times loss(Y_{m2}, Y_f)], \end{aligned} \quad (2)$$

where $Y_1$ and $Y_2$ denote output patches from $x$-$s$ and $y$-$t$ sub-networks respectively, $Y_f$ is the output patch from the fusion module, $Y_{m1}$ and $Y_{m2}$ denote two targets generated from spatial random masks. The gradients of $Y_1$ and $Y_2$ were truncated during the backward projection process. The total loss function of the LF-denoising network is described as the sum of the loss functions of the two sub-networks and the loss function of the fusion module. The parameters of the Adam optimizer were set to $\beta_1 = 0.9$, $\beta_2 = 0.999$. The learning rate was initialized to $1 \times 10^{-4}$ and then decreased by a factor of 0.3 for every 5 steps.

LF-denoising demonstrated its effectiveness across various types of light field data and exhibited robustness in accommodating different input sizes. The network accepted arbitrary angular numbers, provided they were greater than 8. All positional embedding vectors, attention heads and convolution kernels that involve the angular dimension were not hardcoded. Consequently, the parameters of these modules were automatically obtained during the network training stage. For the inference stage, the input light field images were either retained or padded to 81 or 16 (typical angle number used in LFM systems) views, as in the training stage. They were then spatially divided into overlapping EPI patches and rearranged into two forms of EPIs, with pixel sizes of $128 \times 81 \times 128$ or $128 \times 16 \times 128$. The patches were directly passed through the sub-networks and fusion module without undergoing any splitting. The output light field images were angularly padded back to the original input size in the spatial-angular domain. Finally, the output LFs were employed to obtain high-resolution volumes through iterative tomography with DAO, as detailed in previous work[25,51]. The validity of LF-denoising has been verified in extensive fluorescence specimens, including fixed cells, beating hearts of zebrafish larvae, zebrafish embryos, living mouse livers, brains and *Drosophila* brains. To verify the generalization capability of LF-denoising, we trained a network model on bubble and tubulin data and subsequently performed direct inference on cell data without any additional retraining (Supplementary Fig. 11).

The network was implemented on PyTorch platform using a single NVIDIA RTX 3080 GPU. The complete training process for 25 epochs on a typical training set (comprising approximately 1000 pairs) took approximately 5 h. Inference on a light field data with 81 angles (size of $81 \times 459 \times 459$ pixels) required approximately 10.3 s and 2142 MB memory usage. It is worth noting that training and inference speed can be further improved with more powerful GPUs. We have released LF-denoising codes and corresponding 3D reconstruction scripts on a public repository to promote the community.

## Comparison to state-of-the-art denoising methods

We compared our method with previous state-of-the-art denoising methods, such as BM3D, Noise2Void, Noise2Noise, SN2N, DeepInterporation, DeepCAD-RT, DeepSeMi, SRDTrans and V2V3D.

For comparison with BM3D, Noise2Void, Noise2Noise and SN2N, we initially retained the most central 81 angular views in each light field data. Subsequently, we flattened the two angular dimensions into one, rearranging the data into the 3D form of angle × height × width. Following that, we concatenated all rearranged data along the angle dimension, resulting in a 2D image sequence with the length of $81 \times n$, where $n$ denotes the number of light field data. For BM3D, we directly input the 2D images in sequence using codes described previously[52]. For Noise2Noise, Noise2Void and SN2N, each 2D image was cropped into patches with a size of $128 \times 128$ pixels, which was used for network training and inference. The resulting output patches were then stitched back to the full size. The output sequence was rearranged into the form of angle × height × width for subsequent comparison and reconstruction.

For DeepInterpolation, DeepCAD-RT, DeepSeMi and SRDTrans, we also initially retained 81 angular views in each light field data, rearranged them into a 3D form as described above. For static sample datasets without temporal information (Fig. 2 and Supplementary Figs. 5–7), considering the similarity and correlations among angular views, we designated the angle dimension as the temporal dimension for network implementations, resulting in 81-frame-long data sequences, with the form of angle × height × width. For dynamic sample datasets (Figs. 3–6 and Supplementary Fig. 8), we concatenated 3D representations in temporal order for each angular view, yielding 81 image sequences in the form of time × height × width. Each of them was considered as individual data for the networks. For DeepInterpolation, the sequences were cropped into patches with a size of $32 \times 64 \times 64$ pixels, which were used for both training and inference. For DeepCAD-RT and SRDTrans, the sequences were cropped by built-in preprocessing modules with a patch size of $16 \times 128 \times 128$. In the case of DeepSeMi, the temporal patch size was set to 17 as required by the network. After denoising, the output sequence was rearranged into the form of angle × height × width for subsequent comparison and reconstruction. For V2V3D, we retained the most central 13 angular views in each light field data and flattened two angular dimensions into one, rearranging the data into the 3D form of angle × height × width. Subsequently, each 3D image was cropped into patches with a size of $13 \times 128 \times 128$ pixels with overlaps of $16 \times 16$ pixels in spatial dimensions, which was used for network training and inference. The resulting output patches were then stitched back laterally to the full size for subsequent comparison.

All training and inference times were measured on a single NVIDIA RTX 3080 GPU. All the inference times were measured with a light field data with 81 angles (size of $81 \times 459 \times 459$ pixels). The inference time of BM3D was 193.9 s, using 341 MB of memory. The training convergence times for Noise2Void, Noise2Noise, and SN2N were approximately 5, 6, and 4 h, respectively. Their inference times were 170.8, 12.0, and 4.3 s, with memory usage of 1402, 1240, and 1422 MB, respectively. For DeepInterpolation, DeepCAD-RT, DeepSeMi, and SRDTrans, the training convergence times were approximately 5, 2, 7, and 4 h, respectively. Their inference times were 20.2, 6.2, 17.8, and 9.8 s, with memory usage of 3410, 2922, 4620, and 1744 MB, respectively. For V2V3D, with a light field data with 13 angles (size of $13 \times 459 \times 459$ pixels), the training convergence time was approximately 50 h. The average inference time was 32 s, with memory usage of 13,926 MB.

## 3D reconstruction process

After performing denoising on the spatial-angular images, we subsequently reconstruct a 3D volume using an iterative tomography approach[25], which operates in the phase-space domain[51]. A uniform volume is employed as the initial estimate at the start of the iterative

process. During each volume update, different angular components are used sequentially to progressively refine the high-resolution 3D reconstruction. This process involves forward projections, used to estimate the reconstruction error, and backward projections, used to correct the volume accordingly.

## Brain slice preparation

Male Thy1-YFP-H transgenic mice (Jackson stock no. 003782) were transcranially perfused with 50 mL of 0.01 M PBS, followed by 25 mL of 4% PFA in 0.01 M PBS. The harvested brain was fixed in 4% PFA overnight at 4 °C. Brain slices (50 μm thick) were obtained using a Leica VT1200 S vibratome. The slices were then sealed in antifade solution (C1210, Applygen Technologies, Inc.) for high-SNR and low-SNR imaging.

## Zebrafish experiments

Heart beating imaging. *Tg(flk:EGFP*; *gata1:DsRed)* transgenic zebrafish larvae at 4 days postfertilization were used. The larvae were anesthetized by ethyl 3-aminobenzoate methanesulfonate salt (100 mg/L) and mounted in 1% low-melting-point agarose in a 35-mm confocal dish (D35-14-0-N, In Vitro Scientific) for imaging. Embryo imaging. Zebrafish embryos were injected with 300 pg of PH-Mcherry mRNA (synthesized in vitro with an mMessage mMachine T7 kit (AM1344, Ambion)) in one cell at the 16–32 cell stage (1.5 h postfertilization, hpf). At the 70% epiboly stage (8 hpf), injected embryos were embedded in 1% low-melting-point agarose in glass-bottom dishes (D35-14-0-N, In Vitro Scientific) for imaging. Fertilized zebrafish embryos were maintained at 28.5 °C in Holtfreter's solution (NaCl 59 mM, KCl 0.67 mM, CaCl$_2$ 0.76 mM, NaHCO$_3$ 2.4 mM).

## Mouse experiments

Mouse liver imaging. Male wild-type (WT) C57BL/6J mice aged 6–8 weeks were housed with food and water ad libitum under a 12 h light/dark cycle at 24 °C ambient temperature and 50% humidity. 3 μg of WGA-AF488 dye (Cat# W11261, Thermo Fisher) and 1 μg of Alexa Fluor 647 Ly6G antibody (Cat# 127610, Lot# B420110, Clone 1A8, 0.5 mg/ml, Biolegend) and 80 μL of PBS were administered to the mice through tail intravenous injection. Then, Avertin (350 mg/kg, i.p.) was used to anesthetize the mouse deeply by intraperitoneal injection. After 20 min, the mice were dissected to expose the living liver on a homemade holder with a 170 μm-thick glass bottom for imaging. Mouse brain imaging. Male Ai148D mice aged 7–8 weeks were housed with water and food ad libitum under a 12 h light/dark cycle at 24 °C ambient temperature and 50% humidity. A 7 mm craniotomy was made in the mice with an implanted glass window. After 2 weeks of recovery, the mice were head-fixed for imaging.

## *Drosophila* experiments

*Drosophila* was of the genotype: w; UAS-rGRAB_ACh-0.5/+; nSyb-Gal4, UAS-jGCaMP7f, combined by w; UAS-rGRAB_ACh-0.5/cyo; +/(TM2/TM6B), nSyb-Gal4 and UAS-jGCaMP7f. *Drosophila* with the genotype: w; UAS-rGRAB_ACh-0.5/cyo; +/(TM2/TM6B) were from Professor Yulong Li's Lab at Peking University. *Drosophila* with the genotype: nSyb-Gal4 (BDSC: 51941) and UAS-jGCaMP7f were from Professor Yi Zhong's Lab at Tsinghua University. Only the jGCaMP7f channel was analyzed in our experiments. The *Drosophila* were reared on standard cornmeal medium with a 12 h light and 12 h dark cycle, maintained at 23 °C and 60% humidity. The flies were housed in vials containing both male and female individuals, but only female *Drosophila* aged between 3 and 8 days old were selected for this experiment. *Drosophila* were anesthetized using ice and then mounted in a 3D-printed plastic disk, which facilitated unrestricted movement of their legs[53]. Subsequently, the posterior head cuticles, fat bodies and air sacs of the flies were carefully dissected using sharp forceps (5SF, Dumont) at room temperature. The dissection was performed in fresh saline containing

103 mM NaCl, 3 mM KCl, 5 mM TES, 1.5 mM CaCl$_2$, 4 mM MgCl$_2$, 26 mM NaHCO$_3$, 1 mM NaH$_2$PO$_4$, 8 mM trehalose, and 10 mM glucose, adjusted to pH 7.2, bubbled with a mixture of 95% O$_2$ and 5% CO$_2$. Furthermore, we adjusted the positions and angles of the *Drosophila* to maintain the posterior aspect of the head in a horizontal orientation. Additionally, a spacious and clear window was provided for convenient observation of multiple brain regions. Brain movements were minimized by encircling the proboscis with UV glue[54]. Finally, the *Drosophila* were placed under the objective of 2pSAM scanning system for imaging. For neural recording of responses to odor stimulus, 3-octanol, 4-methylcyclohexanol and ethyl acetate (EA), diluted 1.5:1000, 1:1000 and 1:1000, respectively, in mineral oil, were utilized[53]. Odors were released for 5 s with 30 s interval between stimuli, totaling 75 trials (25 sessions), facilitated by a custom-made air pump. Presentation order was pseudo-randomized, and consecutive presentations of the same odor were avoided.

## Performance metrics

We used modulation transfer function (MTF), RMSE, structural similarity (SSIM), PSNR, SNR and Pearson correlation metrics to compare the performance of LF-denoising with existing state-of-the-art methods. The MTF is calculated as

$$MTF = \frac{\max(L) - \min(L)}{\max(L) + \min(L)}, \tag{3}$$

where $L$ is an intensity profile, $\max(L)$ is the peak value, and $\min(L)$ is the trough value. The RMSE is represented as

$$RMSE = \sqrt{\sum_{i=1}^{n} \frac{\|X_i - Y_i\|_2^2}{n}}, \tag{4}$$

where $X$ is the reference, $Y$ is the estimation, $n$ is the pixel number in $X$ or $Y$. The SSIM is represented as

$$SSIM = \frac{(2\mu_X\mu_Y + (0.01 \cdot \max(X))^2)(2\sigma_{XY} + (0.03 \cdot \max(X))^2)}{(\mu_X^2 + \mu_Y^2 + (0.01 \cdot \max(X))^2)(\sigma_X^2 + \sigma_Y^2 + (0.03 \cdot \max(X))^2)}, \tag{5}$$

$\mu_X$ and $\mu_Y$ are the averages of $X$ and $Y$, $\sigma_X$ and $\sigma_Y$, $\sigma_{XY}$ are the corresponding standard deviations and the cross covariance. We firstly obtained the local SSIM maps using sliding 2D Gaussian windows (11 × 11, standard deviation of 1.5) for 2D images and sliding 3D Gaussian windows (11 × 11 × 11, standard deviation 1.5) for 3D volumes. Then the average of these local maps was returned as the final SSIM value. The PSNR is represented as

$$PSNR = 10\log_{10}\frac{\max(\|X\|_2^2)}{\|X - Y\|_2^2}. \tag{6}$$

The SNR is represented as

$$SNR = 10\log_{10}\frac{\|X\|_2^2}{\|X - Y\|_2^2}. \tag{7}$$

The Pearson correlation is represented as

$$R = \frac{E[(X - \mu_X)(Y - \mu_Y)]}{\sigma_X \sigma_Y}, \tag{8}$$

where $R$ is the correlation value, $E[\cdot]$ is the expectation.

## Data analysis

All data processing and analysis were accomplished with customized Python (3.7.0 version) and MATLAB (MathWorks, MATLAB 2019a)

scripts. The 3D rendering was performed by Amira (Thermo Fisher Scientific, Amira 2019) with Voltex modules. The 3D tracking of blood cells in the zebrafish beating heart was carried out automatically using Imaris. For calcium analysis of the mouse brain data, we manually selected regions of interest to acquire temporal traces. The functional traces were calculated using $\Delta F/F_0 = (F-F_0)/F_0$, where $F_0$ represents the average fluorescence within the region of interest averaged over time and $F$ represents the averaged intensity of the region of interest. We further analyzed the spikes of each temporal trace. To filter out local maxima caused by noise, the spikes were identified as the local peaks if they rose surpassing 1% in 3 frames preceding the peaks, were highest peaks in 3 frames within a range of 3 frames before and 10 frames after, and had prominences exceeding 2.5%. All identified local peaks were temporally aligned to the moment their peak value of $\Delta F/F_0$ occurred and normalized, ensuring a minimum value of 0. Subsequently, we averaged all aligned and normalized local peaks within each frame to generate averaged peaks for the raw data and each denoising method employed. Additionally, we calculated the width between the peak rise and fall to 75% of the maximum for each identified local peak.

For calcium analysis of the *Drosophila* data, we manually plotted neural traces of three odor stimulus. Initially, we applied a brain-region mask that selectively included the voxels within the community containing olfactory regions. Subsequently, we extracted masked voxels within 14 frames after 75 trials and reduced dimensionality of voxels using principal component analysis (PCA). In addition, we applied linear discriminant analysis (LDA) on the PCA processed voxels. The third-frame data after the start of odor stimulus were analyzed to identify a low-dimensional space where order identity was stable and well-represented. Following this, 75 trials were categorized into three groups based on the odor released during each trial. Finally, we averaged each group by frame, yielding three latent features shown in Fig. 6h. It is worth noting that, theoretically, the start points of the three latent features should be identical. This is because the olfactory neurons had not yet been stimulated at the time of odor release. Therefore, there should be no statistical differences among all latent features at the start point, despite these features being formed by different odor stimuli. However, in practice, the start points are separated due to neural activity and imaging noises, which would even be further disrupted by DeepCAD-RT and SRDTrans. Furthermore, we partitioned the field of view into 6 horizontal regions. Each region underwent dimensional reduction using the PCA algorithm with the same parameters, resulting in temporal sequences. Following this, we conducted Granger causality tests[55] with a maximum lag of 8 frames (approximately 3.5 s, exceeding the typical neuronal transmission time) between the 6 sequences from frame 2095 to 2185 to produce a Granger causation matrix shown in Fig. 6i. Additionally, we manually selected 3 olfactory neurons (Fig. 6j) and extracted functional traces from frame 6500 to 8000. We then conducted Granger causality tests with the lag of 42 and adaptive thresholds for F-statistic[56] between these 3 traces. We compared the Granger causality matrices produced from raw data and denoising data. We observed that some causal significances present in the raw data were lost after DeepCAD-RT and SRDTrans denoising, while LF-denoising preserved all significant casual relationships and revealed additional causalities compared to raw data (Fig. 6k).

### Ethics statement
This work was conducted with all relevant ethical regulations for animal research. All biological experiments were performed with ethical approval from the Animal Care and Use Committee of Tsinghua University.

### Reporting summary
Further information on research design is available in the Nature Portfolio Reporting Summary linked to this article.

## Data availability
The training data generated in this study have been deposited in the Zenodo database under accession code https://doi.org/10.5281/zenodo.16938668[57]. All data generated in this study are provided in the Source Data file. Source data are provided with this paper.

## Code availability
All relevant codes of LF-denoising are available on GitHub (https://github.com/LF-denoising/LF-denoising) or Zenodo (https://doi.org/10.5281/zenodo.16938935)[58].

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

## Acknowledgements

This work was supported by Natural Science Foundation of China (62088102 Q.D., 62222508 J.W., 62231018 J.W., 62575154 Z.L.), Beijing Natural Science Foundation (Z240011 J.W.), and the Young Scientists Fund of Beijing Natural Science Foundation (4254114 Z.L.).

## Author contributions

Q.D., J.W. and Z.L. conceived and designed the project. Z.L. and F.S. built the optical systems and performed hardware synchronization. W.C., Z.L. and X.L. derived the network architecture. Z.L. and J.W. designed the simulation and biological experiments. W.C., Z.F. and M.J. conducted simulation. Z.L. conducted most biological experiments and data collection. J.F. performed the *Drosophila* experiment and data collection. W.C. and Z.L. conducted data processing and volume reconstructions. Z.L., W.C. and M.J. designed and performed the visualizations. Z.L., W.C., J.W. and Q.D. prepared figures and wrote the manuscript with input from all authors.

## Competing interests

Q.D. and J.W. are founders and equity holders of Zhejiang Hehu Technology, which commercializes related techniques. Q.D., W.C., Z.L. and J.W. submitted patent applications related to the LF-denoising technology described in this work. The remaining authors declare no competing interests.
