## [Transparent Peer Review file · Nature Communications]

Leveraging spatial-angular redundancy for self-supervised denoising of 3D fluorescence imaging without temporal dependency

Corresponding Author: Professor Qionghai Dai

Version 0:

Reviewer comments:

Reviewer #1

(Remarks to the Author)

The manuscript presents an algorithm named LF-denoising, which leverages spatial-angular redundancy in light field microscopy for self-supervised image denoising. The method employs bidirectional angular traversal and EPI-based representation, avoiding the dependence on temporal or spatial redundancy in traditional denoising methods, making it suitable for long-term, low-phototoxicity imaging of highly dynamic biological samples. The method is sound, and the imaging results are rather beautiful. However, several areas require further elaboration to strengthen the scientific contribution of this work:

Major:

1. It appears that LF-denoising maintains spatial resolution significantly better compared to other temporal redundancy denoising methods. In my view, temporal redundancy methods (such as DeepCAD-RT and DeepInterpolation) should primarily affect temporal resolution, potentially causing temporal blurring or event confusion, rather than reducing spatial resolution or angular information. However, the presented results show considerable spatial resolution loss with these methods, as clearly illustrated in Figures 2 and 3 (particularly noticeable in the video results) and indicated by the extremely low MTF values reported in Figure 2c (e.g., DeepInterpolation at 0.2890). I would appreciate further clarification on why this occurs.
2. Denoising before and after light-field tomographic reconstruction is a crucial consideration. I wonder whether the current comparisons in this manuscript have taken this factor into account.
3. The manuscript compares with general denoising methods (BM3D, Noise2Void, DeepCAD-RT, SRDTrans, etc.) but lacks comparison with methods tailored for LFM, such as "V2V3D (CVPR, 2025)". These more targeted comparisons are crucial for validating the advantages of the proposed method.
4. While training and inference times are mentioned, there is no comparison of computational complexity and memory requirements with other methods, which is unfavorable for evaluating practical application feasibility.

Minor:

1. Text repetition in line 149 ("contaminate signals contaminates the signals") needs correction.
2. Lines 368-369 mention that the convolution block attention module contains "3 channel attention layers and 3 spatial attention layers," but the specific implementation details should be provided.
3. Lines 119-120 mention "81 central angular views were utilized to achieve desirable denoising performance," and the Granger causality test in lines 555-556 uses "a maximum lag of 8 frames." The rationale for these parameter choices is recommended to be explained.
4. The paper mentions reconstruction steps in multiple places but does not clearly state the exact position and implementation details of the reconstruction algorithm in the overall workflow, which is important for understanding and

reproducing the results.

(Remarks on code availability)

The code is well-organized and accompanied by a comprehensive README file with clear installation and usage instructions, demonstrating the reproducibility and accessibility of the results for the community.

Reviewer #2

(Remarks to the Author)

1. Quantitative Evaluation of Fixed-Pattern Noise Removal:

In the zebrafish embryo observation experiment (Fig. 4), the proposed method demonstrates effective removal of time-invariant fixed-pattern noise, which is a key advantage of LF-denoising over temporal-based approaches that struggle with such artifacts. However, the evaluation was conducted solely on real-world data without ground truth, limiting the ability to quantitatively assess the performance improvement. To strengthen this claim, I suggest the authors include a simulation-based experiment using commonly used test samples (e.g., a USAF resolution chart) where synthetic fixed-pattern noise can be introduced and the denoising performance quantitatively evaluated against the known ground truth.

2. Clarification of Statistical Significance Indicators:

The statistical significance indicators (asterisks) in Fig. 2d are presented but not clearly explained in the main text or figure caption. It is important to specify which groups were compared for each asterisk to ensure proper interpretation. Similar issues exist in SI Figs. 3 and 5. The authors should clarify these comparisons and consider including a brief explanation in the Methods or figure captions.

3. Provision of Underlying Data for Transparency:

To enhance the reproducibility and transparency of the results, I encourage the authors to provide the numerical data underlying the graphical representations (e.g., box plots, line graphs) in the form of Excel spreadsheets or other machine-readable formats. This will allow readers and reviewers to independently verify the findings and perform further analyses if needed.

Minor Issues:

+Missing Axis Labels in Video 1:

In Video 1, two orthogonal projections are shown from different viewing directions; however, the corresponding axis labels or orientation indicators are missing. Adding these would help viewers better understand the spatial relationships being displayed.

+Unclear Legends in Fig. 1d–e:

The scatter plot legends in Figs. 1d and 1e lack clarity, potentially causing confusion regarding which color corresponds to which method. The authors should consider enlarging the legend markers and explicitly labeling each method to improve readability.

+Need for Clarification of Traces in Fig. 3i:

The description of the traces shown in Fig. 3i is insufficient. The figure legend should clearly explain what each trace represents (e.g., individual neuron activity, averaged response, etc.).

+Missing Unit and Values in SI Fig. 11:

The unit for PSNR in SI Fig. 11 is not indicated. For better quantitative interpretation, the authors should specify the unit (e.g., dB) and include exact numerical values on the bars.

+Missing Scale Bars in SI Fig. 11a–d:

Scale bars are missing from the zoom-in regions in SI Figs. 11a–d. Including them would help readers accurately interpret the spatial dimensions of the displayed images.

+Inconsistent Line Widths in SI Figs. 6 and 7:

In SI Figs. 6 and 7, the line widths used in the legends do not match those in the actual plots. The legend lines appear too thin and may be difficult to distinguish. Consistent and more visible line widths should be used for improved clarity.

+Clarification Needed for “Sample Textures” (Line 115):

In line 115, the phrase “sample textures” could be ambiguous. To avoid confusion, it might be clearer to specify “sample textures in the training set.”

+Clarification of “18 M Parameters” (Line 116):

In line 116, the term “18 M parameters” should be clarified as “18 million parameters” for consistency and readability, especially for a broad audience.

(Remarks on code availability)

I have visited the provided link and confirmed that the datasets and codes for both training and testing are publicly accessible. The authors have also included detailed instructions for running the code, which enhances the reproducibility of the work.

Reviewer #3

(Remarks to the Author)

The authors claim to introduce a new method to denoise shot-noise limited light field microscopy data, leveraging the

spatial-angular correlations in the datasets, which is previously unreported. . The presented results appear impressive in comparison to the methods tested in this paper. The protocol is benchmarked against some of the existing unsupervised denoising methods which demonstrates its efficacy over others. The relevance of the algorithm is shown by testing it over a variety of datasets. The paper is generally well-written, however, some terminology when referring to simulated data is rather confusing. It is unclear how easy the method would be to use in practice for a non-specialist in image processing. Nonetheless, the work is thorough, and should be in principle useful to users of light field microscopy.

There are several minor points that I feel the authors should address to clarify the manuscript.

1) Introduction on light field microscopy could be somewhat expanded. How it compares to confocal, multiphoton, and light-sheet for example (is there commercial availability of LF systems, how widespread it is etc.)

2) The authors talk about photon noise, but do not include the actual intensity values (and bit depth) from the real experimental data. Normalized intensity values are not helpful for comparing with actual data from the lab.

3) Line 379, page 11. "usually set to 0.4" why. Can you expand more on this. Does it mean that for each new dataset this value needs to be empirically evaluated?

4) To include a broader audience, it should be explained if 16/81 views is typical in LFM experiments.

5) For a broad audience, the application of MTF measurement to the data should be expanded. For example, a biologist might not be familiar how this is used, and might be confused about where is the "sinusoidal pattern" in the data.

6) Page 23, Fig 2 caption. Line 775 mentions white arrows. It took me a while to notice them, they should clarify that it is shown in the left most "raw" data image.

7) The authors should emphasise which data is simulated in the figures, not just the captions. For example, Figure 2 uses "raw" to describe simulated data. This is rather confusing.

8) Figures with real, experimental time should include exposure time and power/unit area.

9) Y axis is missing in Figure 3h.

10) The authors should expand how the image data of tubulins and bubbles was simulated (this could be done in the supplementary). What I mean here, is how were the shapes/intensity coded. As this is a useful tool, perhaps putting the code used to simulate on github in jupyter notebook (if python was used) would benefit broader audience.

11) In addition to point 10. Does the simulation of data refer to the fact that the study has been carried out on the artificial Poissonian noise which is signal dependent and so the process ignores all hardware related noise ? Do they use any original datasets, where the dark current noise, fixed pattern noise or the quantization noise come into play ?

12) In the network architecture, spatial random masks are applied to produce the downsampled patches for the fusion module. However, it is not mentioned what is the rationale for this. Are these masks consistent across different datasets?

13) How were the 81 central angular view selected ? Is this an empirical choice or based on other constraints ?

14) Can splitting the two EPIs (in the first stage of the algorithm) amount to loss of angular continuity (and hence the spatial-angular correlations) ? Is there any constraint of how this splitting is done so as to maintain their contextual relationship or the algorithm is agnostic to this ?

15) There has been a growing use of the unsupervised diffusion models for denoising. How does the proposed method compare to that ?

16) The authors should address the following potentially related work: <https://doi.org/10.1117/12.2666000> and <https://doi.org/10.1109/CVPRW.2012.6239346>

Please note that the github repository is not accessible.

(Remarks on code availability)
Repository is not accessible.

Reviewer #4

(Remarks to the Author)
I co-reviewed this manuscript with one of the reviewers who provided the listed reports. This is part of the Nature

Communications initiative to facilitate training in peer review and to provide appropriate recognition for Early Career Researchers who co-review manuscripts.

(Remarks on code availability)

Version 1:

Reviewer comments:

Reviewer #1

(Remarks to the Author)

The authors' revisions have addressed many of the initial concerns regarding performance and implementation. I now recommend publication, pending addressing of the following remaining suggestions:

1. The current title is slightly long. A more concise and impactful alternative might be: "Leveraging spatial-angular redundancy for self-supervised denoising in 3D fluorescence imaging." This version retains the core message while improving readability and flow.
2. The explanation provided for why temporal methods fail on highly dynamic samples is logical and well-reasoned, but the argument would be more compelling with quantitative evidence.
3. I suggest adding a controlled simulation where a key performance metric (e.g., SSIM) is plotted against systematically varied sample velocities. Comparing LF-denoising with a temporal approach, such as DeepCAD could clearly show at what point performance diverges. This "tipping point" analysis would visually and quantitatively validate the strength of your method under challenging motion scenarios.
4. The Discussion section would benefit from a brief reflection on the boundaries of the proposed approach. For example, how might performance be affected by a limited number of angular views or by severe, uncorrected optical aberrations? Such an assessment would demonstrate the authors' awareness of practical constraints and inspire further research.
5. The roles of Loss 1, Loss 2, and Loss 3 remain unclear in the current figure captions. Adding a brief description of each—such as whether it supports sub-network supervision or global fusion—would make the figures more self-contained and informative.
6. The use of a hybrid U-Net/Transformer architecture is interesting, but the rationale behind this design choice should be briefly explained. Highlighting why a hybrid model is superior to purely convolutional or purely transformer-based approaches would add depth and help readers appreciate the architectural decisions.
7. In the Data Analysis section (Line 613), please correct "Granger casualty tests" to "Granger causality tests."

(Remarks on code availability)

The code is well-structured and supported by clear documentation.

Reviewer #2

(Remarks to the Author)

I read the revised manuscript carefully. The author has comprehensively addressed the issues I raised, and now I have no other concerns. I recommend this article for publication.

(Remarks on code availability)

I have run the pretrained model with the test data. The results are reasonable.

Reviewer #3

(Remarks to the Author)

We thank the reviewers for addressing our comments and questions. We have no further questions or comments.

(Remarks on code availability)

Submitting as a co-review.

Reviewer #4

(Remarks to the Author)

(Remarks on code availability)

Attachment: Point-by-point responses to reviewers' comments

Reviewer #1:

The manuscript presents an algorithm named LF-denoising, which leverages spatial-angular redundancy in light field microscopy for self-supervised image denoising. The method employs bidirectional angular traversal and EPI-based representation, avoiding the dependence on temporal or spatial redundancy in traditional denoising methods, making it suitable for long-term, low-phototoxicity imaging of highly dynamic biological samples. The method is sound, and the imaging results are rather beautiful. However, several areas require further elaboration to strengthen the scientific contribution of this work:

Response: We appreciate the reviewer for the time and efforts in the review. We thank the reviewer for recognizing that leveraging spatial-angular redundancy is a key advantage of LF-denoising. In the revision, we have added a new experiment comparing LF-denoising with V2V3D, and provided clearer presentations to better highlight the advantages of LF-denoising. The specific modifications are as follows:

Major:

1. It appears that LF-denoising maintains spatial resolution significantly better compared to other temporal redundancy denoising methods. In my view, temporal redundancy methods (such as DeepCAD-RT and DeepInterpolation) should primarily affect temporal resolution, potentially causing temporal blurring or event confusion, rather than reducing spatial resolution or angular information. However, the presented results show considerable spatial resolution loss with these methods, as clearly illustrated in Figures 2 and 3 (particularly noticeable in the video results) and indicated by the extremely low MTF values reported in Figure 2c (e.g., DeepInterpolation at 0.2890). I would appreciate further clarification on why this occurs.

Response: We thank the reviewer for the comment. Temporal redundancy methods rely on the assumption that signals from consecutive frames are nearly identical, while noise is considered to be independent across frames. However, temporal dependency can lead to motion blur when imaging highly dynamic 3D samples. Therefore, in Fig. 3, the rapid beating of the heart causes significant differences between signals from consecutive frames, resulting in a decline in resolution. In Fig. 2, we demonstrate the denoising of single-frame light field images. To apply temporal redundancy methods in this context, we transformed the angular dimension into a temporal dimension for network training. Due to the substantial variation across different angular views in light field images, the signals corresponding to consecutive frames in this transformed space also become dissimilar, which similarly leads to a reduction in resolution. We have added corresponding explanations in the revised manuscript to more clearly articulate the distinction between LF-denoising and previous temporal redundancy methods.

2. Denoising before and after light-field tomographic reconstruction is a crucial consideration. I wonder whether the current comparisons in this manuscript have taken this factor into account.

Response: We thank the reviewer for the comment. We fully agree with the reviewer that denoising before and after light-field tomographic reconstruction is a crucial consideration. Therefore, we conducted a simulation experiment to analyze the effectiveness of denoising at different stages of the reconstruction pipeline. As shown in Fig. R1 (Supplementary Fig. 1 in the revised version), we observed that methods such as DeepInterpolation and SRDTrans are ineffective when applied after reconstruction, with substantial residual noise remaining in the images. This is primarily because

deconvolution alters the distribution of both signal and noise and fails to account for spatial-angular correlations, leading to suboptimal performance. To address this issue, we designed a new denoising method, termed LF-denoising, which is specifically tailored for denoising spatial-angular images prior to reconstruction, rather than post-reconstruction. Since our approach explicitly exploits spatial-angular redundancy, an attribute that no longer exists after reconstruction, it is not meaningful to directly compare the performance of our method before and after reconstruction.

Figure R1 (Supplementary Figure 1 in the revised version) | Denoising performance on volumetric data after 3D reconstruction. **a**, Simulated tubulins and bubbles with severe noise. **b-c**, Denoising results using DeepInterpolation and SRDTrans. **d**, The corresponding ground truth. **e-g**, Error maps of different methods compare to ground truth, highlighting substantial noise residue. Scale bars, 10 μm .

3. The manuscript compares with general denoising methods (BM3D, Noise2Void, DeepCAD-RT, SRDTrans, etc.) but lacks comparison with methods tailored for LFM, such as "V2V3D (CVPR, 2025)". These more targeted comparisons are crucial for validating the advantages of the proposed method.

Response: We thank the reviewer for the comment. V2V3D (Zhao, J., et al., CVPR, 2025) is a recent proposed method for denoising and reconstructing LFM. We have added a simulated experiment to validate the performances of V2V3D and LF-denoising in removing noise from LFM captured data under low-photon conditions. As shown in Fig. R2, V2V3D restored most high-intensity structures but failed to recover low intensity details, particularly linear structures. In contrast, LF-denoising preserves more details both visually and quantitatively.

V2V3D employs a view2view-based framework for self-supervised denoising, which relies only on angular redundancy and struggled to preserve low-intensity spatial details. LF-denoising leverages spatial-angular redundancy to better distinguish noise-corrupted structures from actual noise, enabling it to preserve more spatial details while maintaining high fidelity. Moreover, V2V3D depends on light-field tomographic reconstruction, which requires substantially greater computational resources compared with denoising directly from light-field measurements. We have added the corresponding results in the revised manuscript.

Figure R2 (Supplementary Figure 13 in the revised version) | Comparison between V2V3D and LF-denoising on the data captured by sLFM. a, MIPs of raw data, ground truth and enhancements by V2V3D and LF-denoising after reconstruction. b-d, Comparison of raw-data and enhancements by V2V3D and LF-denoising in terms of PSNR (b), SSIM (c) and Pearson correlation (d). Scale bars, 10 μm (original view) and 5 μm (enlarged view).

4. While training and inference times are mentioned, there is no comparison of computational complexity and memory requirements with other methods, which is unfavorable for evaluating practical application feasibility.

Response: We thank the reviewer for the comment. We have compared the training time, inference time and memory requirements of LF-denoising, with the following state-of-the-art denoising methods: BM3D, Noise2Void, Noise2Noise, SN2N, DeepInterpolation, DeepCAD-RT, DeepSeMi, and SRDTrans. Since BM3D does not require training on GPUs, its training time is not applicable, and its memory requirement refers to random-access memory (RAM) usage. For all other methods, memory requirements were measured based on video random-access memory (VRAM) usage, as this is the primary constraint in practical applications. All training and inference times were measured on a single NVIDIA RTX 3080 GPU. All the inference times were measured with a light field data with 81 angles (size of $81 \times 459 \times 459$ pixels). The average inference time of BM3D was 193.9 seconds, using 341 MB of memory. The training convergence times for Noise2Void, Noise2Noise, and SN2N were approximately 5, 6, and 4 hours, respectively. Their average inference times were 170.8, 12.0, and 4.3 seconds, with memory usage of 1,402, 1,240, and 1,422 MB, respectively. For DeepInterpolation, DeepCAD-RT, DeepSeMi, and SRDTrans, the training convergence times were approximately 5, 2, 7, and 4 hours, respectively. Their inference times were 20.2, 6.2, 17.8, and 9.8 seconds, with memory usage of 3,410, 2,922, 4,620, and 1,744 MB, respectively. For V2V3D, the training convergence time was approximately 50 hours. The average inference time was 32 seconds, with memory usage of 13,926 MB. The proposed LF-denoising method required approximately 5 hours for training convergence, with an inference time of 10.3 seconds and memory usage of 2,142 MB. We have added the corresponding descriptions in the revised manuscript.

Minor:

1. Text repetition in line 149 ("contaminate signals contaminates the signals") needs correction.

Response: We thank the reviewer for the comment. We have corrected this typo in the revised manuscript.

2. Lines 368-369 mention that the convolution block attention module contains "3 channel attention layers and 3 spatial attention layers," but the specific implementation details should be provided.

Response: We thank the reviewer for the comment. The channel attention layers first calculated the spatial max pooling and spatial average pooling of the patches. These two pooled representations were then processed by a double convolution layer with shared weights. Finally, the output of channel attention was obtained by adding two convoluted features. The spatial attention layers first computed the channel-wise average and max of the patch while preserving the spatial dimensions. These two results were concatenated into a two-channel feature and then reduced into a single channel using a convolution layer. It is worth noting that the outputs of both channel attention layers and spatial attention layers were element-wise multiplied with the input features to produce the final output features. We have added the corresponding descriptions in the revised manuscript.

3. Lines 119-120 mention "81 central angular views were utilized to achieve desirable denoising performance," and the Granger causality test in lines 555-556 uses "a maximum lag of 8 frames." The rationale for these parameter choices is recommended to be explained.

Response: We thank the reviewer for the comment. In our practical LFM system, there are totally 169 angular views available. However, using all of them would significantly slow down both network training and inference. To improve computational efficiency, we utilized only 81 central angular views. As shown in Fig. R3 (Supplementary Fig. 4 in the revised version), we quantitatively observed that using 81 central angular views accelerates training by a factor of 2.7 and inference by a factor of 1.7, while maintaining denoising performance nearly equivalent to that achieved when using all 169 angular views.

Figure R3 (Supplementary Figure 4 in the revised version) | Angle selection strategy for LF-denoising. **a**, Reconstruction results for high-SNR input data, using all 81 angles (left) and 169 central angles (right). **b**, Reconstruction results for low-SNR input data after LF-denoising, using 81 angles (left) and 169 angles (right). **c**, Strategy of bidirectional angular traversals to establish EPI data pairs, for 81 angles (top) and 169 angles (bottom). **d**, Convergence curves of LF-denoising with different number of angles used for training. **e**, PSNR metrics for reconstructions of high-SNR input data (left) and low-SNR input data after LF-denoising (right), using 169 angles and 81 angle. All P values were calculated using paired *t*-test, significance at $P < 0.05$. * $P < 0.05$, ** $P < 0.01$, *** $P < 0.001$. $n = 12$ individual samples. P values are 9.88×10^{-1} and 4.76×10^{-2} from left to right. **f**, Comparison of processing time for LF-denoising using different number of angles. Scale bars, 10 μm (**a-b**).

For the Granger causality test, the transmission of calcium signals between neurons occurs very rapidly, typically within less than 3 seconds. The neural calcium signals were recorded at a sampling rate of 2.3 volumes per second (VPS). Therefore, we selected a maximum lag corresponding to 8 frames (approximately 3.5 seconds), which exceeds the typical neuronal transmission time, to ensure that relevant causal relationships could be captured. We have added corresponding descriptions in the revised manuscript.

4. The paper mentions reconstruction steps in multiple places but does not clearly state the exact position and implementation details of the reconstruction algorithm in the overall workflow, which is important for understanding and reproducing the results.

Response: We thank the reviewer for the comment. After performing denoising on the spatial-angular images, we subsequently reconstruct a 3D volume using an iterative tomography approach (Wu, J. et al, *Cell*, 184, 3318-3332, 2021), which operates in the phase-space domain (Lu, Z. et al, *Optics Express*, 27, 18131-18145, 2019). A uniform volume is employed as the initial estimate at the start of the iterative process. During each volume update, different angular components are used sequentially to progressively refine the high-resolution 3D reconstruction. This process involves forward projections, used to estimate the reconstruction error, and backward projections, used to correct the volume accordingly. We have added a new subsection in the Methods to describe the reconstruction process.

Remarks on code availability:

The code is well-organized and accompanied by a comprehensive README file with clear installation and usage instructions, demonstrating the reproducibility and accessibility of the results for the community.

Response: We thank the reviewer for the comment. We expect that LF-denoising will be widely used in biological observations.

Reviewer #2:

1. Quantitative Evaluation of Fixed-Pattern Noise Removal:

In the zebrafish embryo observation experiment (Fig. 4), the proposed method demonstrates effective removal of time-invariant fixed-pattern noise, which is a key advantage of LF-denoising over temporal-based approaches that struggle with such artifacts. However, the evaluation was conducted solely on real-world data without ground truth, limiting the ability to quantitatively assess the performance improvement. To strengthen this claim, I suggest the authors include a simulation-

based experiment using commonly used test samples (e.g., a USAF resolution chart) where synthetic fixed-pattern noise can be introduced and the denoising performance quantitatively evaluated against the known ground truth.

Response: We thank the reviewer for the constructive suggestion. We have added a simulation experiment to validate the abilities of LF-denoising and DeepCAD-RT to remove fixed-pattern noise on USAF resolution chart, as shown in Fig. R4 (Supplementary Figure 9 in the revised version). The simulated noise consists of time-invariant fixed pattern noise and time-variant gaussian noise. The result shows that DeepCAD-RT fails to remove time-invariant fixed pattern noise, whereas LF-denoising removes both types of noise and preserves detailed structures both visually and quantitatively. We have added the results in the revised version.

Figure R4 (Supplementary Figure 9 in the revised version) | Fix pattern noise removing comparison between DeepCAD-RT and LF-denoising. **a**, Center view measurements of raw data, ground truth and enhancements by DeepCAD-RT and LF-denoising. **b-d**, Comparison of raw-data and enhancements by DeepCAD-RT and LF-denoising in terms of PSNR (**b**), SSIM (**c**) and Pearson correlation (**d**). Scale bars, 500 μm (original view) and 100 μm (enlarged view).

2. Clarification of Statistical Significance Indicators:

The statistical significance indicators (asterisks) in Fig. 2d are presented but not clearly explained in the main text or figure caption. It is important to specify which groups were compared for each asterisk to ensure proper interpretation. Similar issues exist in SI Figs. 3 and 5. The authors should clarify these comparisons and consider including a brief explanation in the Methods or figure captions.

Response: We are sorry that our previous illustration is unclear. For the benchmarks in Fig. 2d, Supplementary Figs. 3 and 5, the asterisks indicate the significant levels of comparisons between each method and LF-denoising. We have added the corresponding descriptions, as shown in Fig. R5 (Fig. 2d), Fig. R6 (Supplementary Fig. 3g), and Fig. R7 (Supplementary Fig. 5).

Fig. R5 (Fig. 2d in the revised version) | Boxplots showing comparisons between different denoising methods using RMSE (lower is better), SSIM, PSNR and Pearson correlation (higher is better). Metrics are calculated on data after reconstruction. The boxplot formats: center line, median; box limits, lower and upper quartiles; whiskers, 1.5-fold interquartile range. All P values were calculated using paired *t*-test, significance at $P < 0.05$. * $P < 0.05$, ** $P < 0.01$, *** $P < 0.001$. $n = 10$ denotes the number of individual samples. P values are 1.71×10^{-19} , 9.03×10^{-19} , 7.35×10^{-25} , 3.55×10^{-23} , 2.71×10^{-15} , 3.51×10^{-24} and 1.28×10^{-11} from left to right for RMSE, and 2.95×10^{-19} , 9.84×10^{-19} , 6.98×10^{-33} , 2.48×10^{-17} , 4.72×10^{-23} , 2.38×10^{-28} and 2.49×10^{-12} from left to right for SSIM, and 9.02×10^{-25} , 1.35×10^{-22} , 6.85×10^{-39} , 3.02×10^{-27} , 8.39×10^{-16} , 3.29×10^{-35} and 3.43×10^{-17} from left to right for PSNR, and 3.76×10^{-18} , 2.51×10^{-18} , 1.89×10^{-22} , 1.63×10^{-23} , 8.43×10^{-18} , 5.41×10^{-25} and 1.33×10^{-10} from left to right for Pearson correlation. The asterisks indicate the significant levels of comparisons between each method and LF-denoising.

Fig. R6 (Supplementary Fig. 3g in the revised version) | Boxplot showing PSNR and Pearson correlation indices for different methods. The boxplot formats: center line, median; box limits, lower and upper quartiles; whiskers, 1.5-fold interquartile range. All P values were calculated using paired *t*-test, significance at $P < 0.05$. * $P < 0.05$, ** $P < 0.01$, *** $P < 0.001$. $n = 10$ individual samples. P values are 2.47×10^{-7} , 8.43×10^{-3} ,

1.89×10^{-3} and 2.41×10^{-3} from left to right for PNSR, and 5.35×10^{-9} , 1.01×10^{-3} , 8.13×10^{-5} and 2.17×10^{-5} from left to right for Pearson correlation. The asterisks indicate the significant levels of comparisons between each method and LF-denoising.

Fig. R7 (Supplementary Fig. 5 in the revised version) | Benchmark of LF-denoising against other methods before and after reconstruction. **a**, Example angular views of simulated tubulins and bubbles captured by sLFM with severe noise are shown in the left column. Image bit depth is set to 16, variance of Gaussian noise to 5, and photon number of the maximum intensity to 10. Results processed by BM3D, noise2void, noise2noise, DeepInterpolation, DeepCAD-RT, DeepSeMi, SRDTrans and our LF-denoising are shown in the middle columns. Right column shows the ground truth without noise. Boxplots compares different methods in terms of RMSE, SSIM, PSNR, Pearson correlation. The boxplot formats: center line, median; box limits, lower and upper quartiles; whiskers, 1.5-fold interquartile range. All P values were calculated using paired

t-test, significance at $P < 0.05$. * $P < 0.05$, ** $P < 0.01$, *** $P < 0.001$. $n = 10$ individual samples. P values are 1.75×10^{-17} , 8.99×10^{-17} , 5.96×10^{-14} , 1.69×10^{-14} , 6.53×10^{-15} , 7.09×10^{-11} , 1.36×10^{-14} , 1.44×10^{-14} , 4.81×10^{-19} , 7.79×10^{-21} , 5.75×10^{-18} , 3.45×10^{-18} , 9.04×10^{-17} , 1.90×10^{-15} , 1.90×10^{-15} , 5.62×10^{-16} , 1.95×10^{-25} , 6.38×10^{-24} , 2.66×10^{-15} , 3.28×10^{-16} , 1.26×10^{-19} , 9.64×10^{-12} , 6.74×10^{-21} , 4.66×10^{-16} , 1.11×10^{-28} , 1.34×10^{-17} , 8.90×10^{-19} , 3.36×10^{-19} , 3.05×10^{-20} , 1.22×10^{-17} , 2.67×10^{-22} and 7.53×10^{-20} from left to right for metric evaluation. The asterisks indicate the significant levels of comparisons between each method and LF-denoising. **b**, Corresponding results after reconstruction, displayed as orthogonal MIPs. P values are 1.12×10^{-18} , 6.70×10^{-21} , 5.98×10^{-14} , 3.91×10^{-14} , 2.85×10^{-20} , 2.08×10^{-12} , 2.16×10^{-18} , 3.33×10^{-14} , 1.02×10^{-24} , 6.22×10^{-23} , 3.24×10^{-16} , 5.50×10^{-18} , 4.02×10^{-25} , 4.43×10^{-17} , 3.00×10^{-20} , 3.75×10^{-17} , 4.58×10^{-19} , 2.54×10^{-18} , 1.33×10^{-13} , 4.99×10^{-13} , 1.09×10^{-18} , 4.34×10^{-12} , 1.63×10^{-16} , 5.05×10^{-14} , 3.89×10^{-29} , 4.46×10^{-31} , 1.09×10^{-14} , 1.27×10^{-23} , 1.27×10^{-32} , 7.52×10^{-21} , 1.44×10^{-28} and 5.63×10^{-20} from left to right. The asterisks indicate the significant levels of comparisons between each method and LF-denoising. Scale bars, 20 μm (original view) and 5 μm (enlarged view).

3. Provision of Underlying Data for Transparency:

To enhance the reproducibility and transparency of the results, I encourage the authors to provide the numerical data underlying the graphical representations (e.g., box plots, line graphs) in the form of Excel spreadsheets or other machine-readable formats. This will allow readers and reviewers to independently verify the findings and perform further analyses if needed.

Response: We thank the reviewer for the valuable suggestion aimed at increasing transparency and reproducibility of our results. We have provided the underlying numerical data for all graphical representations in the figures in Excel format.

Minor problems:

1. Missing Axis Labels in Video 1:

In Video 1, two orthogonal projections are shown from different viewing directions; however, the corresponding axis labels or orientation indicators are missing. Adding these would help viewers better understand the spatial relationships being displayed.

Response: We thank the reviewer for the comment. We have added the missing axis instructions in the updated Supplementary Video 1.

2. Unclear Legends in Fig. 1d–e: The scatter plot legends in Figs. 1d and 1e lack clarity, potentially causing confusion regarding which color corresponds to which method. The authors should consider enlarging the legend markers and explicitly labeling each method to improve readability.

Response: We thank the reviewer for the comment. We have added legends with enlarged scatters with method names for further clarification, as shown in Fig. R8 (Figs. 1d-e in the revised version).

Fig. R8 (Figs. 1d-e in the revised version) | LF-denoising performance indicated by the SSIM across varying photon numbers (d) and Gaussian standard deviations (e), with distinct improvement over state-of-the-art methods. Image bit depth is set to 16.

3. Need for Clarification of Traces in Fig. 3i: The description of the traces shown in Fig. 3i is insufficient. The figure legend should clearly explain what each trace represents (e.g., individual neuron activity, averaged response, etc.).

Response: We thank the reviewer for the comment. Each trace represents the spatial position change at the center of each cell, and different colors represent different types of cells that have been identified. We have added further instructions in the figure legend.

Fig. R9 (Supplementary Fig. 12 in the revised version) | Comparison between LF-denoising and SN2N. a-d, Angular measurements of raw data in a, ground truth in b, and enhancements by SN2N in c and LF-denoising in d. e-g, Comparison of raw data and enhancements by SN2N and LF-denoising in terms of PSNR in e, Pearson correlation in f, and SSIM in g.

f and SSIM in **g**. Scale bars, 5 μm .

4. Missing Unit and Values in SI Fig. 11: The unit for PSNR in SI Fig. 11 is not indicated. For better quantitative interpretation, the authors should specify the unit (e.g., dB) and include exact numerical values on the bars.

Response: We thank the reviewer for the comment. We have added the unit “dB” and specific values in Fig. R9 (Supplementary Fig. 12 in the revised version).

5. Missing Scale Bars in SI Fig. 11a–d: Scale bars are missing from the zoom-in regions in SI Figs. 11a–d. Including them would help readers accurately interpret the spatial dimensions of the displayed images.

Response: We thank the reviewer for the comment. The scalebars for zoom-in regions in Fig. R9 (Supplementary Fig. 12 in the revised version) has been added.

6. Inconsistent Line Widths in SI Figs. 6 and 7: In SI Figs. 6 and 7, the line widths used in the legends do not match those in the actual plots. The legend lines appear too thin and may be difficult to distinguish. Consistent and more visible line widths should be used for improved clarity.

Response: We thank the reviewer for the comment. We have bolded the linewidths for legends in Supplementary Figs. 6 and 7, as shown in Fig. R10 (Supplementary Fig. 6c in the revised version) and Fig R11 (Supplementary Fig. 7c in the revised version).

Fig. R10 (Supplementary Fig. 6c in the revised version) | Curves of SSIM and Pearson correlation versus varying photon numbers applied for different methods.

Fig. R11 (Supplementary Fig. 7c in the revised version) | Curves of SSIM and Pearson correlation versus varying Gaussian standard deviations applied for different methods.

7. Clarification Needed for “Sample Textures” (Line 115): In line 115, the phrase “sample textures” could be ambiguous. To avoid confusion, it might be clearer to specify “sample textures in the training set.”

Response: We thank the reviewer for the comment. We have modified this expression in the revised manuscript.

8. Clarification of “18 M Parameters” (Line 116): In line 116, the term “18 M parameters” should be clarified as “18 million parameters” for consistency and readability, especially for a broad audience.

Response: We thank the reviewer for the comment. We have modified this expression in the revised manuscript.

Remarks on code availability:

I have visited the provided link and confirmed that the datasets and codes for both training and testing are publicly accessible. The authors have also included detailed instructions for running the code, which enhances the reproducibility of the work.

Response: We thank the reviewer for the comment. We expect that LF-denoising will be widely used in biological observations.

Reviewer #3:

The authors claim to introduce a new method to denoise shot-noise limited light field microscopy data, leveraging the spatial-angular correlations in the datasets, which is previously unreported. The presented results appear impressive in comparison to the methods tested in this paper. The protocol is benchmarked against some of the existing unsupervised denoising methods which demonstrates its efficacy over others. The relevance of the algorithm is shown by testing it over a variety of datasets. The paper is generally well-written, however, some terminology when referring to simulated data is rather confusing. It is unclear how easy the method would be to use in practice for a non-specialist in image processing. Nonetheless, the work is thorough, and should be in principle

useful to users of light field microscopy.

Response: We thank the reviewer for the positive comments to illustrate the advantages of LF-denoising in speed and fidelity. We have clarified the terminology when referring to simulated data, to facilitate the method to be used in practice for a non-specialist in image processing. The specific modifications are as follows:

There are several minor points that I feel the authors should address to clarify the manuscript.

1) Introduction on light field microscopy could be somewhat expanded. How it compares to confocal, multiphoton, and light-sheet for example (is there commercial availability of LF systems, how widespread it is etc.)

Response: We thank the reviewer for the comment. The core advantage of LFM lies in its highly parallelized 3D volumetric imaging capability. Traditional confocal (Pawley, J. Springer Science & Business Media, 2006) and two-photon microscopes (Denk, W. et al, *Science*, 248, 73-76, 1990) require slow, point-by-point scanning across the 3D volume to achieve volumetric imaging. More recently, light-sheet microscopy (Stelzer, E. H. et al, *Nature Reviews Methods Primers*, 1, 73, 2021) has improved the physical scanning mechanism, yet it still relies on sequential plane-by-plane scanning, which remains relatively slow and is further constrained by sample transparency. In contrast, LFM leverages high-dimensional spatial-angular information to enable single-shot 3D volumetric imaging with a single camera exposure. The unique capability has greatly advanced large-scale, high-speed biological applications and is now becoming commercially available and widely used by the scientific community (An, H. et al, *Science Advances*, 11, eadq6399, 2025; Chen, S. et al, *Advanced Science*, 11, 2306066, 2024; Xiang, Q. et al, *American Journal of Respiratory Cell and Molecular Biology*, 2025). We have added corresponding descriptions in the Introduction section of the revised manuscript.

2) The authors talk about photon noise, but do not include the actual intensity values (and bit depth) from the real experimental data. Normalized intensity values are not helpful for comparing with actual data from the lab.

Response: We thank the reviewer for the comment. Normalized intensity is primarily used in Figs. 3h, 4e, and 6g to present the comparison between different methods in terms of their ability to resolve adjacent blood cells, trends of photobleaching, and neural response traces. Normalization allows for clearer comparison across methods by minimizing the influence of absolute intensity variations.

That said, we fully agree with the reviewer that the actual intensity values and bit depth from the experimental data are important. Therefore, we have provided the actual intensity values before normalization in the Source Data Files. All experimental data were acquired at a bit depth of 16 bits.

3) Line 379, page 11. “usually set to 0.4” why. Can you expand more on this. Does it mean that for each new dataset this value needs to be empirically evaluated?

Response: We thank the reviewer for the comment. In our experiments, the weight values were fixed at 0.4 and does not need to be empirically evaluated for each new dataset. To avoid ambiguity, we have explicitly set the weight value to 0.4 in the formula of the loss function, as shown below:

$$\begin{aligned} loss_{fusion} = & 0.4 \times \left[0.5 \times loss(Y_1, Y_f) + 0.5 \times loss(Y_2, Y_f) \right] \\ & + 0.6 \times \left[0.5 \times loss(Y_{m1}, Y_f) + 0.5 \times loss(Y_{m2}, Y_f) \right], \end{aligned}$$

where Y_1 and Y_2 denote output patches from x - s and y - t sub-networks respectively, Y_f is the output patch from the fusion module, Y_{m1} and Y_{m2} denote two targets generated from spatial random masks. The gradients of Y_1 and Y_2 were truncated during the backward projection process. We have made corresponding modifications in the revised manuscript.

4) To include a broader audience, it should be explained if 16/81 views is typical in LFM experiments.

Response: We thank the reviewer for the comment. In different LFM configurations, the number of angular views may vary, as it is determined by the intrinsic design and optical configuration of the system. Using 16 or 81 angular views is very typical in LFM setups. We have added corresponding descriptions in the revised manuscript.

5) For a broad audience, the application of MTF measurement to the data should be expanded. For example, a biologist might not be familiar how this is used, and might be confused about where is the “sinusoidal pattern” in the data.

Response: We thank the reviewer for the comment. We used the modulation transfer function (MTF) measurement to evaluate the capability of different methods to resolve structures that are closely spaced. In Fig. R12 (Fig. 2c in the revised version), we provide the MTF measurement of the ground truth image. Therefore, the closer the MTF measurement of a given method is to that of the ground truth image, the better the performance of that method. In the case of methods such as BM3D, due to their limited resolving capability, no distinct valley appears in the middle of the MTF curve, which indicates inferior performance compared to LF-denoising. We have added corresponding descriptions in the revised manuscript.

Fig. R12 (Fig. 2c in the revised version) | Another enlarged regions after reconstruction, with insets showing corresponding Fourier transforms. The normalized intensity profiles and modulation transfer functions (MTFs) of white arrows in the left most image across adjacent tiny structure are shown for comparison. The presence of a valley in the middle of the MTF curve indicates strong resolving capability of LF-denoising for spatially adjacent structures.

6) Page 23, Fig 2 caption. Line 775 mentions white arrows. It took me a while to notice them, they should clarify that it is shown in the left most “raw” data image.

Response: We thank the reviewer for the comment. We have modified the figure legend, as the reviewer requested.

7) The authors should emphasise which data is simulated in the figures, not just the captions. For

example, Figure 2 uses “raw” to describe simulated data. This is rather confusing.

Response: We thank the reviewer for the comment. We sincerely apologize for any confusion caused. As illustrated in Fig. R13 (Fig. 2 in the revised version), the term "raw" in the original Fig. 2 has been replaced with "simulated noisy image".

Fig. R13 (Fig. 2 in the revised version) | Benchmark of LF-denoising with various denoising methods on the data captured by LFM. a, Center views of AICS cells labelled with microtubules, imaged using sLFM at low SNR in simulation, before and after LF-denoising. The image bit depth is set to 16, Gaussian noise variance to 5, and the photon number of maximum intensity to 100. **b,** Enlarged regions obtained by different denoising methods, including BM3D, Noise2Void, Noise2Noise, DeepInterpolation, DeepCAD-RT, DeepSeMi, SRDTrans and LF-denoising. Two example angular views are shown. High-SNR data is regarded as ground truth. Yellow and blue arrows indicate the image blur and detail loss. **c,** Another enlarged regions after reconstruction, with insets showing corresponding Fourier transforms. The normalized intensity profiles and modulation transfer functions (MTFs) of white arrows in the left most image across adjacent tiny structure are shown for comparison. The presence of a valley in the middle of the MTF curve indicates strong resolving capability of LF-denoising for spatially adjacent structures. **d,** Boxplots showing comparisons between

different denoising methods using RMSE (lower is better), SSIM, PNSR and Pearson correlation (higher is better). Metrics are calculated on data after reconstruction. The boxplot formats: center line, median; box limits, lower and upper quartiles; whiskers, 1.5-fold interquartile range. All P values were calculated using paired *t*-test, significance at $P < 0.05$. * $P < 0.05$, ** $P < 0.01$, *** $P < 0.001$. $n = 10$ denotes the number of individual samples. P values are 1.71×10^{-19} , 9.03×10^{-19} , 7.35×10^{-25} , 3.55×10^{-23} , 2.71×10^{-15} , 3.51×10^{-24} and 1.28×10^{-11} from left to right for RMSE, and 2.95×10^{-19} , 9.84×10^{-19} , 6.98×10^{-33} , 2.48×10^{-17} , 4.72×10^{-23} , 2.38×10^{-28} and 2.49×10^{-12} from left to right for SSIM, and 9.02×10^{-25} , 1.35×10^{-22} , 6.85×10^{-39} , 3.02×10^{-27} , 8.39×10^{-16} , 3.29×10^{-35} and 3.43×10^{-17} from left to right for PSNR, and 3.76×10^{-18} , 2.51×10^{-18} , 1.89×10^{-22} , 1.63×10^{-23} , 8.43×10^{-18} , 5.41×10^{-25} and 1.33×10^{-10} from left to right for Pearson correlation. The asterisks indicate the significant levels of comparisons between each method and LF-denoising. Scale bars, $5 \mu\text{m}$ (a-c) and $2 \mu\text{m}^{-1}$ (c).

8) Figures with real, experimental time should include exposure time and power/unit area.

Response: We thank the reviewer for the comment. Comprehensive details regarding the actual experiments, including exposure time, average power per unit area, and other relevant parameters, have been provided in Table R1 (Supplementary Table 1).

Table R1 (Supplementary Table 1) | Imaging conditions for all fluorescence experiments

	Sample (imaging T, °C)	Fluorescent label	λ : average power/unit area (mW/mm ²)	Laser exposure time (# time pts)	Volume rate (VPS)	Objective	Setup, angular resolution
3	Zebrafish larva (27 °C)	Tg(flk:EGFP; gata1:DsRed)	488: 1.5 561: 2.5	10 ms 100 pts	50	63×/1.4NA Oil	LFM, inverted, 13 × 13
4	Zebrafish embryo (27 °C)	EGFP	488: 0.01	1 ms 36,940 pts	1	63×/1.4NA Oil	sLFM, inverted, 13 × 13
4d (50× SNR)	Zebrafish embryo (27 °C)	EGFP	488: 0.5	50 ms 36,940 pts	1	63×/1.4NA Oil	sLFM, inverted, 13 × 13
5a-5f	Living mouse liver (37 °C)	Ly6G WGA	488: 0.07 561: 0.08	3 ms 455 pts	1/20	63×/1.4NA Oil	sLFM, inverted, 13 × 13
5a-5f (20× SNR)	Living mouse liver (37 °C)	Ly6G WGA	488: 1.4 561: 1.6	60 ms 455 pts	1/20	63×/1.4NA Oil	sLFM, inverted, 13 × 13
5g-5l	Living mouse brain (37 °C)	GCamp6f	488: 0.01	0.1 ms 1,110 pts	30	25×/1.05N A Water	sLFM, upright, 21 × 21
5g-5l (50× SNR)	Living mouse brain (37 °C)	GCamp6f	488: 0.5	5 ms 1,110 pts	30	25×/1.05N A Water	sLFM, upright, 21 × 21
6	Drosophila brain (27 °C)	jGCaMP7f	920: 5.1	400 ms 6,000 pts	2.3	25×/1.05N A Water	2pSAM, upright, 13
S8 (high SNR)	Mouse brain slice (27 °C)	THy1-YFP	488: 1.0	100 ms 500 pts	4	63×/1.4NA Oil	sLFM, inverted, 13 × 13

S8 (low SNR)	Mouse brain slice (27 °C)	THy1-YFP	488: 0.01	1 ms 500 pts	4	63×/1.4NA Oil	sLFM, inverted, 13 × 13
-----------------	---------------------------------	----------	-----------	-----------------	---	------------------	----------------------------

9) Y axis is missing in Figure 3h.

Response: We thank the reviewer for the comment. We have added the Y axis indicating normalized intensity in Fig. R14 (Fig. 3h in the revised version).

Fig. R14 (Fig. 3h in the revised version) | Normalized intensity profiles along the lines indicated by arrows in g, showing that adjacent cells, which were unresolved by previous algorithms, were distinguished by LF-denoising.

10) The authors should expand how the image data of tubulins and bubbles was simulated (this could be done in the supplementary). What I mean here, is how were the shapes/intensity coded. As this is a useful tool, perhaps putting the code used to simulate on github in jupyter notebook (if python was used) would benefit broader audience.

Response: We thank the reviewer for the comment. The tubulins and bubbles employed in this study were derived from the buhtub dataset published in our previous work (Lu, Z. et al, *Nature Methods*, 2025 <https://doi.org/10.1038/s41592-025-02698-z>). All relevant codes of generating buhtub dataset are readily accessible and available on GitHub (<https://github.com/kimchange/SeReNet>) and Zenodo (<https://doi.org/10.5281/zenodo.14909862>). We have added the corresponding information in the revised manuscript.

11) In addition to point 10. Does the simulation of data refer to the fact that the study has been carried out on the artificial Poissonian noise which is signal dependent and so the process ignores all hardware related noise? Do they use any original datasets, where the dark current noise, fixed pattern noise or the quantization noise come into play?

Response: We thank the reviewer for the comment. In our simulation data, low signal-to-noise ratio (SNR) images were generated by artificially introducing noise. The primary noise types include: 1) the shot noise and dark current noise, governed by Poisson distributions and dependent on photon counts; 2) the readout noise, following a Gaussian distribution parameterized by standard deviation σ . These noise models were incorporated into our simulations and described in corresponding figure legends. For instance, in Fig. 2, simulated noisy images were created by setting the image bit depth is set to 16, Gaussian noise variance to 5, and the photon number of maximum intensity to 100. Supplementary Figs. 6 and 7 demonstrate consistent denoising performance across varying Poisson and Gaussian noise levels. LF-denoising surpasses state-of-the-art methods in noise removal efficacy, confirming its robustness across diverse scenarios involving shot noise, dark current noise, and readout noise.

To further quantify fixed-pattern noise removal, we have added a simulation experiment to validate the capabilities of LF-denoising to remove fixed-pattern noise on USAF resolution chart,

as shown in Fig. R15 (Supplementary Figure 9 in the revised version). The simulated noise consists of time-invariant fixed pattern noise and time-variant gaussian noise. The result shows that previous methods like DeepCAD-RT fail to remove time-invariant fixed pattern noise, whereas LF-denoising removes both types of noise and preserves detailed structures both visually and quantitatively. We have added the results in the revised version.

Figure R15 (Supplementary Figure 9 in the revised version) | Fix pattern noise removing comparison between DeepCAD-RT and LF-denoising. **a**, Center view measurements of raw data, ground truth and enhancements by DeepCAD-RT and LF-denoising. **b-d**, Comparison of raw-data and enhancements by DeepCAD-RT and LF-denoising in terms of PSNR (**b**), SSIM (**c**) and Pearson correlation (**d**). Scale bars, 500 μ m (original view) and 100 μ m (enlarged view).

In addition, regarding the scientific sCMOS cameras we used, quantization noise typically exhibits substantially lower magnitude compared to the shot noise and readout noise, rendering its influence negligible. Consequently, we excluded the quantization noise from our analysis.

12) In the network architecture, spatial random masks are applied to produce the downsampled patches for the fusion module. However, it is not mentioned what is the rationale for this. Are these masks consistent across different datasets?

Response: We thank the reviewer for the comment. The purpose of introducing spatial random masks is to prevent network from overfitting to artefacts generated during spatial downsampling processes. The masks group every four spatial pixels of the input patch and randomly select two of them according to one of eight predefined patterns, resulting in two downsampled patches with half the original width and height. For angular dimension, the masks were identical across different views and always picked the same pixel. These masks were randomly generated each training iteration, and the random strategy is consistent across different datasets. We have added the corresponding descriptions in the revised manuscript.

13) How were the 81 central angular view selected? Is this an empirical choice or based on other

constraints?

Response: We thank the reviewer for the comment. In our practical LFM system, there are totally 169 angular views available. However, using all of them would significantly slow down both network training and inference. To optimize processing throughput without sacrificing denoising efficacy, we strategically adopted a reduced subset of angular views. As shown in Fig. R16 (Supplementary Fig. 4 in the revised version), we quantitatively observed that using 81 central angular views accelerates training by a factor of 2.7 and inference by a factor of 1.7, while maintaining denoising performance nearly equivalent to that achieved when using all 169 angular views. This experimental validation confirms that 81 central views suffice for LF-denoising to faithfully reconstruct clean images, justifying our selection of the angular sampling scheme. We have added the corresponding descriptions in the revised manuscript.

Figure R16 (Supplementary Figure 4 in the revised version) | Angle selection strategy for LF-denoising. **a**, Reconstruction results for high-SNR input data, using all 81 angles (left) and 169 central angles (right). **b**, Reconstruction results for low-SNR input data after LF-denoising, using 81 angles (left) and 169 angles (right). **c**, Strategy of bidirectional angular traversals to establish EPI data pairs, for 81 angles (top) and 169 angles (bottom). **d**, Convergence curves of LF-denoising with different number of angles used for training. **e**, PSNR metrics for reconstructions of high-SNR input data (left) and low-SNR input data after LF-denoising (right), using 169 angles and 81 angle. All P values were calculated using paired *t*-test, significance at $P < 0.05$. * $P < 0.05$, ** $P < 0.01$, *** $P < 0.001$. $n = 12$ individual samples. P values are 9.88×10^{-1} and 4.76×10^{-2} from left to right. **f**, Comparison of processing time for LF-denoising using different number of angles. Scale bars, $10 \mu\text{m}$ (a-b).

14) Can splitting the two EPIs (in the first stage of the algorithm) amount to loss of angular continuity (and hence the spatial-angular correlations)? Is there any constraint of how this splitting is done so as to maintain their contextual relationship or the algorithm is agnostic to this?

Response: As demonstrated in Fig. R17 (Fig. 1b in the revised version), our separation of the two EPIs operates exclusively along the *x* and *y* dimensions, thereby preserving angular continuity. However, this process may potentially compromise spatial continuity. To address this limitation,

we implemented bidirectional constraints through x - s and y - t EPIs, which effectively preserves spatial-angular correlations. As shown in Fig. R18 (Supplementary Fig. 3 in the revised manuscript), it reveals that retaining only the x - s or y - t EPI pathway would disrupt these correlations and consequently degrade resolution during denoising. The bidirectional EPI architecture of LF-denosing successfully resolves this challenge while maintaining reconstruction fidelity. We have added the corresponding descriptions in the revised manuscript.

Figure R17 (Figure 1b in the revised version) | Schematic of the LF-denosing processing pipeline. Noisy spatial-angular images are first converted into two EPIs via bidirectional angular traversing. These noisy EPIs are divided into four sub-stacks along the x and y dimensions, serving as inputs and targets for training transformer-based networks in a self-supervised manner. An attention-based module then fuses the two channels to produce clean spatial-angular images. Multiple orthogonal masks and downsampling operators are employed to establish a global loss function, enhancing fidelity. Scale bars: 20 μm .

Figure R18 (Supplementary Figure 3 in the revised version) | Ablation study on LF-denoising. **a**, Noisy angular measurements. **b**, Ground truth without noise. **c**, Simplified schematic of LF-denoising-1, using only x - s directional EPI for self-supervision. Angular images after LF-denoising-1 are shown. **d**, Simplified schematic of LF-denoising-2, using only y - t directional EPI for self-supervision. Angular images after LF-denoising-2 are shown. **e-f**, Simplified schematic of LF-denoising, using bidirectional EPIs for self-supervision. Data augmentation is used in **f**. Angular images after LF-denoising are shown. **g**, Boxplot showing PSNR and Pearson correlation indices for different methods. The boxplot formats: center line, median; box limits, lower and upper quartiles; whiskers, 1.5-fold interquartile range. All P values were calculated using paired t -test, significance at $P < 0.05$. * $P < 0.05$, ** $P < 0.01$, *** $P < 0.001$. $n = 10$ individual samples. P values are 2.47×10^{-7} , 8.43×10^{-3} , 1.89×10^{-3} and 2.41×10^{-3} from left to right for PSNR, and 5.35×10^{-9} , 1.01×10^{-3} , 8.13×10^{-5} and 2.17×10^{-5} from left to right for Pearson correlation. The asterisks indicate the significant levels of comparisons between each method and LF-denoising. Scale bars, $5 \mu\text{m}$ (**a-f**).

15) There has been a growing use of the unsupervised diffusion models for denoising. How does the proposed method compare to that?

Response: We thank the reviewer for the comment. Unsupervised diffusion models offer significant advantages for image denoising, primarily by eliminating the need for paired clean-noisy training data, leveraging instead the inherent structure within noisy observations themselves, making them particularly valuable for real-world applications (Croitoru, F. et al, *IEEE Transactions on Pattern Analysis and Machine Intelligence*, 45(9), 10850-10869, 2023). However, these benefits come with notable drawbacks. The iterative nature of reverse diffusion necessitates hundreds to thousands of network evaluations per image, making inference orders of magnitude slower than single-pass discriminative models like LF-denoising, severely hindering real-time use. Performance of diffusion models is also highly sensitive to the choice of sampling parameters, requiring careful tuning to avoid artefacts or excessive smoothing. These hyperparameter adjustments pose challenges for biologists. In contrast, the proposed LF-denoising method offers a faster, more stable, and practical solution for denoising microscopy data. We have added the corresponding descriptions in the Discussion section of the revised manuscript.

16) The authors should address the following potentially related work: <https://doi.org/10.1117/12.2666000> and <https://doi.org/10.1109/CVPRW.2012.6239346>

Response: We thank the reviewer for the suggestion. We have added these two references in the revised manuscript.

Remarks on code availability:

Please note that the github repository is not accessible. Repository is not accessible.

Response: We thank the reviewer for the comment. The GitHub repository may be inaccessible due to potential network restrictions. To facilitate access, we have additionally provided a Zenodo repository link (<https://doi.org/10.5281/zenodo.16938935>) to publicly share the relevant codes for LF-denoising.

Attachment: Point-by-point responses to reviewers' comments

Reviewer #1:

The authors' revisions have addressed many of the initial concerns regarding performance and implementation. I now recommend publication, pending addressing of the following remaining suggestions:

Response: We thank the reviewer for the positive comment and efforts in significantly improving the quality of our manuscript. During this revision process, we further supplemented the simulation experiments, quantitative analysis, and descriptive discussion, as requested by the reviewer.

1. The current title is slightly long. A more concise and impactful alternative might be: "Leveraging spatial-angular redundancy for self-supervised denoising in 3D fluorescence imaging." This version retains the core message while improving readability and flow.

Response: We sincerely thank the reviewer for the thoughtful suggestion regarding the title. We fully understand the concern about conciseness and truly appreciate the proposed alternative. After careful consideration, we would like to retain the phrase "without temporal dependency," as it conveys an essential aspect of our work. As noted in the reviewer's comment #3, we have added new experiments demonstrating that traditional methods such as DeepCAD-RT rely on temporal information, whereas our approach remains effective even when sample motion accelerates, avoiding motion blur and preserving denoising performance. Moreover, the experiments shown in Figs. 3-6 were specifically designed to verify that LF-denoising operates independently of temporal information, ranging from handling rapid cardiac motion and removing fixed pattern artefacts to improving imaging fidelity while preserving temporal causality. We therefore hope that retaining this phrase in the title can help highlight this distinguishing feature in a concise and informative way.

2. The explanation provided for why temporal methods fail on highly dynamic samples is logical and well-reasoned, but the argument would be more compelling with quantitative evidence.

Response: We thank the reviewer for the comment. We have added a quantitative analysis on simulated tubulins and bubbles following comment #3. The temporal method DeepCAD-RT failed on highly dynamic samples, both visually and quantitatively. We have added the corresponding results in the revised manuscript.

3. I suggest adding a controlled simulation where a key performance metric (e.g., SSIM) is plotted against systematically varied sample velocities. Comparing LF-denoising with a temporal approach, such as DeepCAD could clearly show at what point performance diverges. This "tipping point" analysis would visually and quantitatively validate the strength of your method under challenging motion scenarios.

Response: We thank the reviewer for the suggestion. We have added a simulated experiment to evaluate the performance of DeepCAD-RT and LF-denoising under varied sample velocities, as shown in Figure R1 (Supplementary Figure 8 in the revised version). The results show that the performance of DeepCAD-RT degraded as the velocity increased, whereas LF-denoising preserved fine structural details without motion blur. This further supports the advantage of LF-denoising that eliminates the dependency on temporal redundancy. We have added the corresponding result in the revised manuscript.

Figure R1 (Supplementary Figure 8 in the revised version) | Comparison between DeepCAD-RT and LF-denoising on varied sample velocities. **a**, Center view measurements of simulated tubulins and bubbles captured by sLFM at increasing sample velocities are shown in the first row. The same sample was translated horizontally from left to right. Image bit depth is set to 16, variance of Gaussian noise to 5, and photon number of the maximum intensity to 10, the imaging speed to 42 Hz. Results processed by DeepCAD-RT and LF-denoising are shown in subsequent rows. **b**, Ground truth without noise. **c**, Curves of SSIM versus varying sample velocities applied for DeepCAD-RT and LF-denoising. Scale bars, 20 μm (a, b).

4. The Discussion section would benefit from a brief reflection on the boundaries of the proposed approach. For example, how might performance be affected by a limited number of angular views or by severe, uncorrected optical aberrations? Such an assessment would demonstrate the authors' awareness of practical constraints and inspire further research.

Response: We thank the reviewer for the suggestion. The performance of LF-denoising was evaluated on 2pSAM data with only 13 angular views. Further reducing angular views would likely degrade performance, as spatial-angular redundancy collapses toward purely spatial redundancy. LF-denoising does not presume any properties of the imaging system, including optical aberrations. The method has been verified to generalize across different spatial-angular imaging setups with varying optical aberrations. Nevertheless, severe aberrations that substantially disrupt angular redundancy may adversely affect the performance of LF-denoising, which can be addressed using computational adaptive optics. We have added these discussions in the revised manuscript.

5. The roles of Loss 1, Loss 2, and Loss 3 remain unclear in the current figure captions. Adding a brief description of each—such as whether it supports sub-network supervision or global fusion—would make the figures more self-contained and informative.

Response: We thank the reviewer for the suggestion. Loss 1 and Loss 2 are sub-network loss functions defined on the EPI representation, whereas Loss 3 is a global loss function defined on fused spatial-angular representation. We have added corresponding descriptions in the figure legend.

6. The use of a hybrid U-Net/Transformer architecture is interesting, but the rationale behind this design choice should be briefly explained. Highlighting why a hybrid model is superior to purely convolutional or purely transformer-based approaches would add depth and help readers appreciate the architectural decisions.

Response: We thank the reviewer for the comment. We designed this hybrid U-Net/Transformer architecture to balance accuracy and efficiency. The U-net convolutional encoders extract compact local features at low cost and reduces the dimensionality. Transformer then captures long-range dependencies without heavy computation at full resolution. Then decoders recover spatial details with skip connections. This hybrid design is more computational efficient than a purely Transformer model and preserves more local structure than a purely convolutional U-Net. We have added corresponding rationale in the revised manuscript.

7. In the Data Analysis section (Line 613), please correct “Granger casualty tests” to “Granger causality tests.”

Response: We thank the reviewer for the comment. We have corrected this typo in the revised manuscript.

The code is well-structured and supported by clear documentation.

Response: We thank the reviewer for the comment. We expect that LF-denoising will be widely used in biological observations.

Reviewer #2:

I read the revised manuscript carefully. The author has comprehensively addressed the issues I raised, and now I have no other concerns. I recommend this article for publication.

I have run the pretrained model with the test data. The results are reasonable.

Response: We highly appreciate your generous words of acknowledging the contributions of our revised manuscript.

Reviewer #3:

Submitting as a co-review.

Response: We are extremely grateful to the co-reviewers for their efforts in evaluating our manuscript.